# Charting differentially methylated regions in cancer with Rocker-meth

Matteo Benelli [1,7✉], Gian Marco Franceschini [2,7], Alberto Magi [3], Dario Romagnoli[1], Chiara Biagioni [1,4], Ilenia Migliaccio[5], Luca Malorni [4,5] & Francesca Demichelis [2,6✉]

Differentially DNA methylated regions (DMRs) inform on the role of epigenetic changes in cancer. We present Rocker-meth, a new computational method exploiting a heterogeneous hidden Markov model to detect DMRs across multiple experimental platforms. Through an extensive comparative study, we first demonstrate Rocker-meth excellent performance on synthetic data. Its application to more than 6,000 methylation profiles across 14 tumor types provides a comprehensive catalog of tumor type-specific and shared DMRs, and agnostically identifies cancer-related partially methylated domains (PMD). In depth integrative analysis including orthogonal omics shows the enhanced ability of Rocker-meth in recapitulating known associations, further uncovering the pan-cancer relationship between DNA hyper-methylation and transcription factor deregulation depending on the baseline chromatin state. Finally, we demonstrate the utility of the catalog for the study of colorectal cancer single-cell DNA-methylation data.

[1] Bioinformatics Unit, Hospital of Prato, Prato, Italy. [2] Department of Cellular, Computational and Integrative Biology, University of Trento, Trento, Italy. [3] Department of Information Engineering, University of Florence, Florence, Italy. [4] "Sandro Pitigliani" Medical Oncology Department, Hospital of Prato, Prato, Italy. [5] "Sandro Pitigliani" Translational Research Unit, Hospital of Prato, Prato, Italy. [6] Institute for Computational Biomedicine, Weill Cornell Medical College, New York, NY, USA. [7] These authors contributed equally: Matteo Benelli, Gian Marco Franceschini. ✉email: matteo.benelli@uslcentro.toscana.it; f.demichelis@unitn.it

DNA methylation is an essential player of gene regulation and therefore one of the most studied epigenetic mechanisms[1–5]. It is considered to be a surrogate of chromatin accessibility and, especially when in the proximity of regulatory regions, alterations of DNA methylation status may correlate positively or negatively to the binding of key transcription factors (TFs), resulting in altered transcriptional programs[6–8]. Alterations of DNA methylation have been associated with a wide range of diseases, including cancer. Landscape integrative studies from The Cancer Genome Atlas (TCGA) consortium have highlighted marked associations between DNA methylation and other genomic layers. For instance, the Prostate Adenocarcinoma (PRAD) group of TCGA reported that unsupervised analysis of DNA methylation levels naturally recapitulated the distinct molecular subclasses of primary prostate cancer, originally defined by the presence of mutually exclusive recurrent molecular alterations including Single Nucleotide Variants (SNVs), Somatic Copy Number Alterations (SCNAs), and gene fusions[9]. Similar results have been observed for multiple tumor types[10–14]. Furthermore, DNA methylation has recently gained substantial attention in the setting of liquid biopsy for early detection and for monitoring of disease evolution[15–18], further eliciting the need for robust and reliable biomarker detection approaches.

To date, the majority of studies on DNA methylation focused on the characterization of single CpG sites, despite the recognition that concomitant changes spanning entire genomic regions, referred to as Differentially Methylated Regions (DMRs), are common in cancer tissues with respect to benign cells[19–21]. The hypo-methylation (compared to matched normal tissue) of large contiguous stretches of repetitive DNA sequences is considered to be a common hallmark of cancer initiation and indicative of chromatin and genomic instability[22,23]. Similarly, tumor cells often exhibit hyper-methylation of promoter CpG islands of tumor suppressor genes that may be associated to their inactivation and consequent tumor cell proliferation[24,25].

Initial studies on the identification of DMRs used strategies based on grouping of Differentially Methylated Sites (DMSs) into a-priori defined regions, such as CpG islands, followed by the application of a statistical test (i.e., a Fisher Exact Test) to define statistically significant DMRs[26–28]. More recently, a series of computational methods were specifically designed to identify DNA regions showing differential methylation status between two phenotypes (i.e., cancer versus adjacent tissue). These include both methods originally developed for microarray data, such as Bumphunter[29] and DMRcate[30] and bisulfite sequencing (BS) based methods, such as DSS[31], DMRseq[32], Metilene[33], and others[34–36]. While a multitude of array-based cancer DNA methylation data, such as TCGA[37,38], can serve as comparative sets, the use of genome-wide and targeted sequencing-based methylation studies is increasing, particularly in the context of low signal to noise ratios as in the patients' circulation. For these reasons, computational tools amenable and benchmarked for both platforms and able to detect DMRs without a-priori constraints neither on the magnitude of the differential signal nor on the spanned genomic size are desirable to allow for the establishment of harmonized catalogs for downstream biological studies.

Here we report on a new computational method to identify DMRs in both array and BS data. We show that our newly developed approach, named Rocker-meth, allowed for the unbiased detection of DMRs spanning a range of genomic sizes, from hundreds of base pairs to Megabase-pair (Mbp). By a comprehensive synthetic study, we demonstrate its superiority compared to state-of-the-art methods. The application of Rocker-meth to whole-genome BS (WGBS) and HM450 array TCGA datasets allowed us to compile an accurate catalog of DMRs, representing the full characterization of differential methylation in 14 TCGA datasets including about 6000 tumor samples. In addition, we show the utility of our catalog in the analysis of single-cell DNA methylation datasets. Rocker-meth recapitulates both well-known and previously underappreciated associations between DNA-methylation and genomic features, gene expression, and chromatin states validating its accuracy and allowing us to refine the functional and regulatory role of DNA methylation in human cancers.

## Results

**Rocker-meth and its application to multiple synthetic and cancer datasets.** Rocker-meth (Receiver operating characteristic curves analyzer of DNA methylation data, Fig. 1a) consists of four main modules: (1) computation of Area Under the Curve (AUC) values from Receiver operating characteristic (ROC) Curve analysis of methylation levels (i.e., beta values) in tumor versus normal samples; CpG sites with AUC value toward either 1 or 0 capture hyper-methylation and hypo-methylation events, respectively; (2) segmentation of AUC values by a tailored heterogeneous Hidden Markov Model (HMM)[39]; (3) estimation of intra-segment homogeneity by Wilcoxon–Mann–Whitney (WMW) test on beta values of CpG sites in tumor versus normal samples; (4) identification of sample specific DMRs by Z-score statistics. Details are reported in Methods section. To showcase Rocker-meth potential (Fig. 1b), we applied it to multiple independent cancer datasets generated using a variety of assays, including HM450 arrays from TCGA, WGBS[40], and single-cell DNA-methylation[41] and further evaluated its capability of capturing biologically relevant features using orthogonal omics data, such as multiple genomic features, matched gene expression data from TCGA, and chromatin states[4,42].

We first carried out a comprehensive simulation study on RRBS synthetic datasets (details in Supplementary Note 1) to evaluate the performance of the segmentation algorithm of Rocker-meth while varying the values of its parameters (details in the Methods section). The results of this analysis are reported in Supplementary Note 1, Supplementary Figs. 1 and 2. Next, to assess its overall performance, we carried out an extensive comparative study on multiple synthetic datasets emulating increasing levels of signal-to-noise ratio (class 1–5) and different platforms (i.e., HM450, RRBS -parameters training, WGBS) (details in Supplementary Data 1 and methods) against five state-of-art-methods, including three sequencing-based tools, Metilene[33], DSS[31] and DMRseq[32], one array-based tool Bumphunter[29], and DMRcate[30] that can be applied to both data types. Supplementary Fig. 4 shows the distribution of the average beta values of tumor versus normal samples for differentially methylated sites in TCGA datasets, estimated as those sites with AUC < 0.2 (hypo-methylation) or AUC > 0.8 (hyper-methylation). For illustration, overlaid bars referring to the range (estimated by mean ± standard deviation) of the beta difference emulated by the different synthetic datasets are reported. Importantly, class 4 and 5 can recapitulate about 34% and 21% of all differential methylated events, respectively, corresponding to a total of 41% (partial overlap between the two classes) (Supplementary Fig. 4 and Supplementary Data 1). We used the precision, recall, and F1 score statistics (i.e., the harmonic mean of precision and recall statistics) to evaluate the performance of the methods and, following the benchmark strategy described in Jühling et al.[33], we evaluated both site-wise and segment-wise statistics. The dot chart reported in Fig. 1c (left) and data reported in Supplementary Data 2 summarize the segment-wise statistical measures of all methods in correctly identifying DMRs for the different synthetic WGBS datasets. While in the high and middle

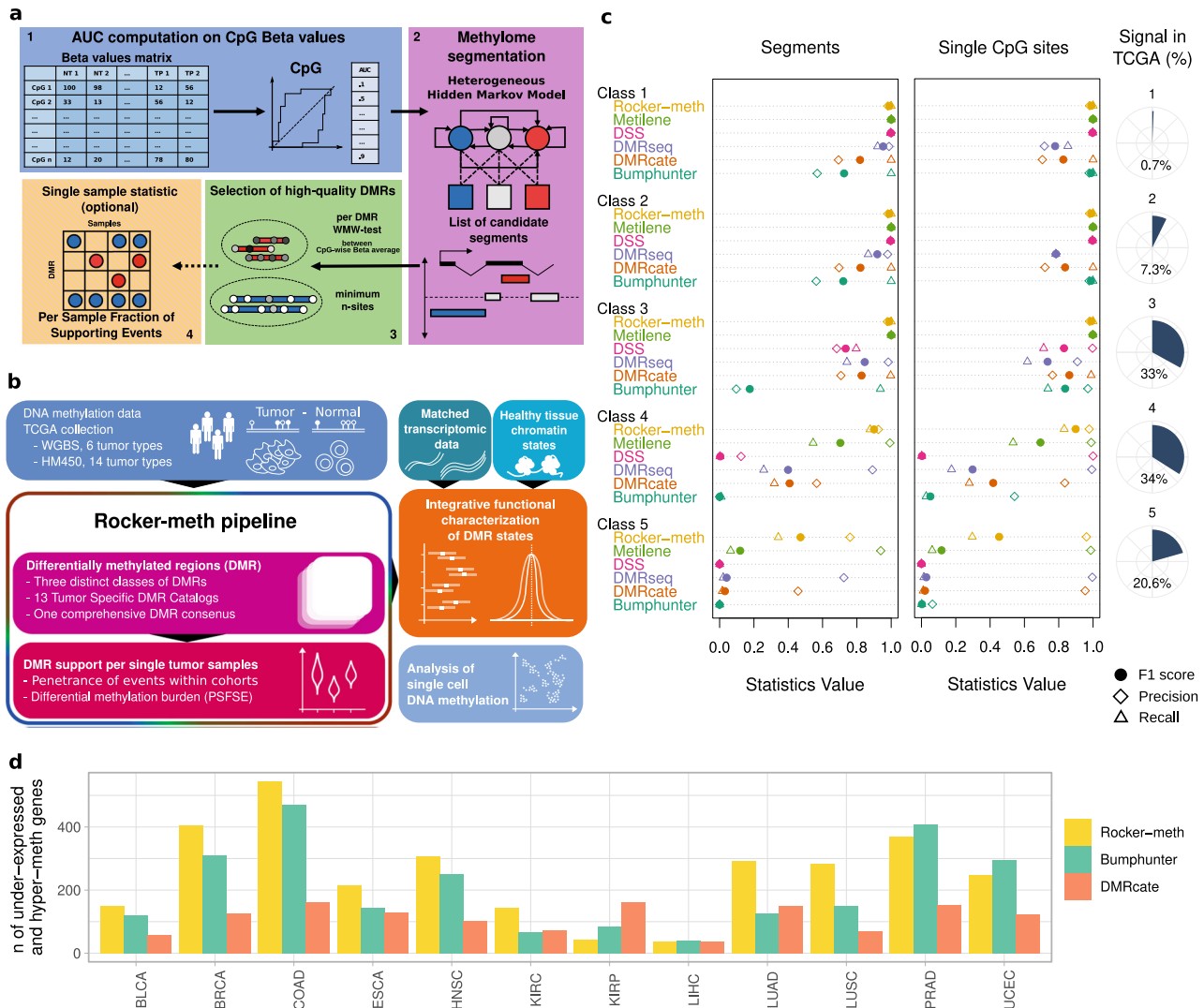

**Fig. 1 Rocker-meth study design and performance on synthetic datasets. a** The Rocker-meth method involves four main steps: (1) computation of Area Under the Curve (AUC) values of methylation levels (i.e., beta values) in tumor versus normal samples; (2) segmentation of AUC values by a tailored heterogeneous Hidden Markov Model (HMM); (3) filters on segments features including intra-segment homogeneity and number of CpG sites; (4) identification of sample specific DMRs by Z-score statistics (optional). **b** To showcase Rocker-meth, we applied it to multiple DNA-methylation cancer datasets, generated using array (HM450) and sequencing (WGBS and single-cell DNA-methylation) data. We also exploited orthogonal omics data (gene expression data from TCGA and ENCODE chromatin states) to evaluate its capability of capturing biologically relevant features. **c** Comparative study in synthetic DNA methylation datasets. Right: Dot chart summarizing segment-wise (left) and site-wise (right) precision (diamonds), recall (triangles) and F1 (filled circles) statistical measures for Bumphunter, DMRcate, DMRseq, DSS, Metilene, and Rocker-meth in the five WGBS synthetic datasets. Right: distribution of the differences of per site percentage of methylation (beta) between tumor and normal samples for all differentially methylated sites across 14 tumor types (TCGA datasets). The range of beta difference classes is defined on synthetic datasets and termed from class 1 (highest signal-to-noise ratio) to class 5 (lowest signal-to-noise ratio). The fraction of emulated TCGA data is reported as pie chart. **d** Bar plots of the number of hyper-DMRs in promoter-TSS, 5′ UTR, or first intron of under-expressed genes. Bars refer to the number of events detected by Rocker-meth, Bumphunter, and DMRcate across the different tumor types. DMR differentially methylated region, TCGA the cancer genome atlas, PSFSE per sample fraction of supporting events, WGBS whole genome bisulfite sequencing, HM450 Illumina 450k array, TP tumor primary, NT normal adjacent tissue.

signal-to-noise ratio datasets (class 1–3), all tools showed good to excellent performance, in the datasets with the lowest signal-to-noise-ratio (classes 4 and 5) Rocker-meth outperformed the other methods, demonstrating markedly higher precision and recall than both sequencing and array-based methods. For all methods the decrease in terms of F1 was due to a decrease in recall statistics (i.e., the ability to detect true positive events). Nonetheless, in class 5 datasets Rocker-meth was still able to identify about 32% of events, followed by Metilene (7.5%), DMRcate (5%), and DMRseq (3%). Similar results were obtained for site-wise statistics, summarized in Fig. 1c (right).

Comparable results related to synthetic HM450 data and the parameters training dataset RRBS are also reported (Supplementary Data 3–4, Supplementary Figs. 5 and 6). In terms of specificity, all methods showed excellent performance. In particular, Rocker-meth obtained specificity >0.99 in all the synthetic datasets. In terms of computational performance, Bumphunter was the most efficient, followed by Metilene and Rocker-meth (Supplementary Fig. 7).

We then assessed the performance of Rocker-meth and the other two array-based methods considered in this study (Bumphunter and DMRcate) in the analysis of real DNA

methylation data on a total of 712 normal and 5623 cancer samples across 14 tumor types from TCGA. As no ground truth exists, we focused on events that are more likely to occur, specifically hyper-methylation in promoter-TSS, 5′ UTR, or first intron and under-expression. Figure 1d (top) shows the number of under-expressed genes with a DMR in/spanning their promoter-TSS, 5′ UTR, or first intron as predicted by Rocker-meth, Bumphunter or DMRcate. We observed that Rocker-meth identified a higher number of these events in 8 out of 12 tumor types compared to Bumphunter and 10 out of 12 tumor types compared to DMRcate. Importantly, we observed that these results were not due to an over-calling of DMRs. In fact, as reported in Supplementary Fig. 8a the number of hyper-DMRs detected by Rocker-meth was always lower or comparable to the number of DMRs identified by Bumphunter or DMRcate. These results clearly suggest that Rocker-meth is able to detect more expected (i.e., based on biological knowledge) events across different tumor types than the other state-of-the-art methods (high sensitivity) while keeping the number of detected events low (high specificity). Next, we focused on these events and run functional enrichment analysis using clusterProfiler[43] on the set of Gene Ontology (GO) terms (C5: ontology gene sets from MSigDB Collections version 7.4). First, we observed that the number of significant (FDR < 0.05) GO terms from Rocker-meth results was markedly higher than those from Bumphunter or DMRcate in 10 out of 12 tumor types (Supplementary Fig. 8b). In addition, Rocker-meth led to more robust GO terms enrichment, as suggested by significantly lower distribution of FDRs in 6 out of 12 tumor types compared to Bumphunter and 10 out of 12

compared to DMRcate (Supplementary Fig. 8c). We then studied the ability of the methods in recapitulating other well-known features of DNA methylation in human cancers. As reported in Supplementary Fig. 8d, compared to Bumphunter and DMRcate Rocker-meth predicts higher enrichment of hyper-DMRs in promoters and 5′ associated with gene under-expression, higher enrichment of hyper-DMRs versus hypo-DMRs in CpG islands, and higher enrichment of hypo-DMRs versus hyper-DMRs in intergenic regions.

**Application of Rocker-meth to 6 WGBS TCGA datasets.** We applied Rocker-meth to WGBS data of 27 cancer samples from 6 tumor types (BLCA, BRCA, COAD, LUAD, LUSC, and UCEC), both per tumor-type and pan-cancer. A median of 127,218 loss-of-methylation and of 16,316 gain-of-methylation regions were identified (Fig. 2a), with a median of 31.4% and 0.71% of the cancer genome (for gain-of-methylation and loss-of-methylation regions, respectively) characterized by differential methylation. The highest DMR burden (i.e, the fraction of cancer genome within DMRs) was observed for BLCA (Fig. 2b), where DMRs span 70.5% of the genome. All tumor types showed similar levels of gain-of-methylation burden, with values ranging from 0.18% to 1.12%. Overall, we observed that loss-of-methylation regions were significantly larger than gain-of-methylation ones (Supplementary Fig. 9, WMW $p$-value $<2.2e-16$); while the distribution of the length of gain-of-methylation regions was not dependent from the genomic annotation (maximum around 10 Kb), loss-of-methylation regions tended to be larger when affecting intergenic regions (Fig. 2c, $p$-value $<2.2e-16$), thus impacting on the

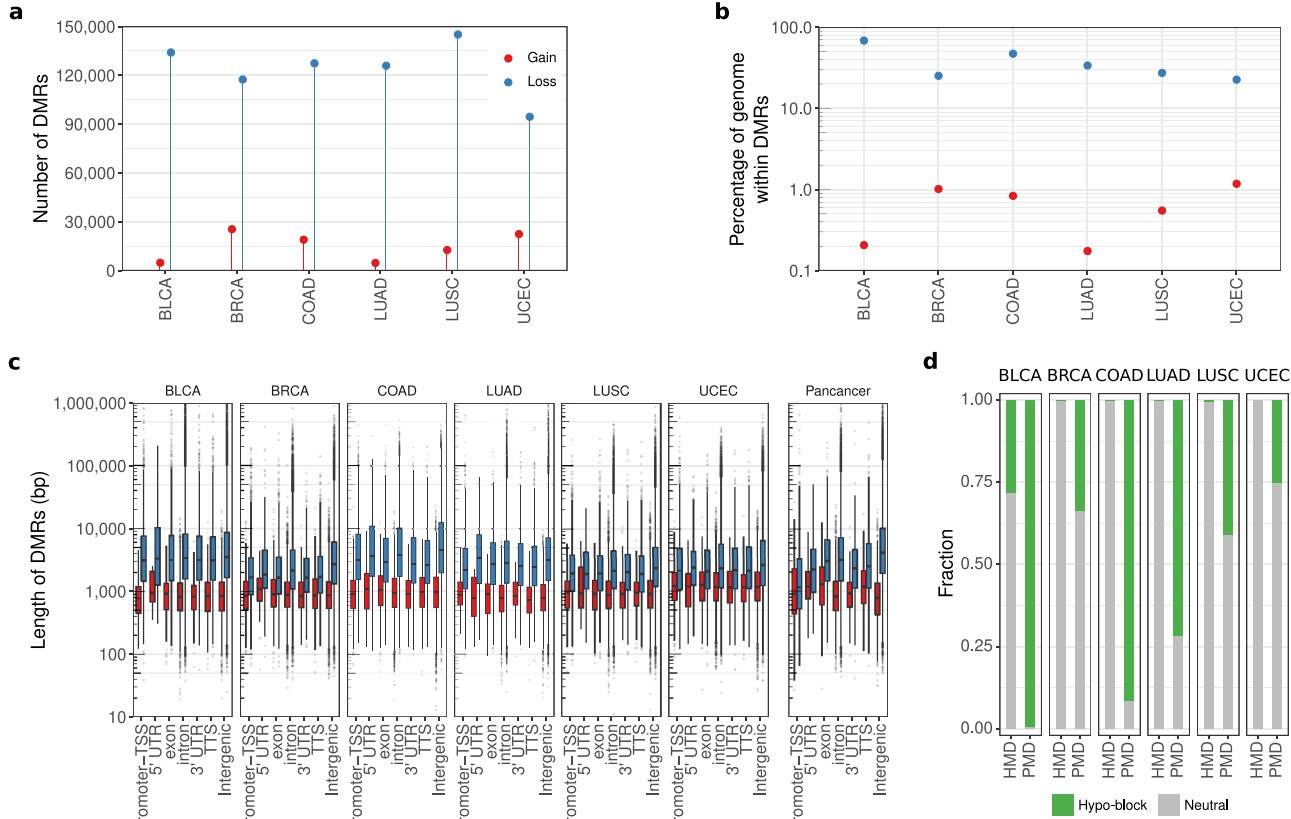

**Fig. 2 Differentially methylated regions from whole genome bisulfite data. a** Lollipop plot showing the number of regions with gain (red) and loss (blue) of methylation detected by Rocker-meth across 6 tumor types from TCGA. **b** Dot plot of the fraction of cancer genome within DMRs (DMR burden) for gain and loss of methylation events across the different tumor types. **c** Box plot reporting the distribution of DMR lengths (base pairs, bp) for gain and loss of methylation events across tumor types and genic features. **d** Fraction of the overlap between partially methylated domains (PMD) and highly methylated domains (HMD) segments and the tumor type-specific hypo-blocks identified by Rocker-meth.

overall size difference. Based on these observations on WGBS data, which homogeneously capture the entire genome space, we implemented the following operative classification for DMR states with respect to the reference normal samples by distinguishing wide from short/modest regions demonstrating loss of methylation: (i) Hyper: a gain-of-methylation region of any size, (ii) Hypo: a loss-of-methylation region with length ≤ 10 Kbp, (iii) hypo-block: a loss-of-methylation region with length > 10 Kbp.

Relevant to this operative distinction, multiple studies showed evidence of large portions of the genome particularly prone to stochastic hypo-methylation, usually termed Partially Methylated Domains (PMDs). These regions are generally intergenic, with genomic sizes as large as 10 Mb[26], and exhibit increased demethylation as a tissue ages, with the most prominent demethylation level in cancer[40,44]. We studied the association between hypo-blocks and a recent catalog of PMDs[40] and found an extremely significant overlap between these two classes of events (Fig. 2d, p-val <2.2e-16). Of note, Highly Methylated Domains (HMD) (i.e., highly methylated regions that are conserved in normal and cancer cells) correspond to regions identified as not-differentially methylated by Rocker-meth, demonstrating the high specificity of our tool on WGBS data. Importantly, Rocker-meth nominates hypo-blocks that are tumor-type specific, further refining the current knowledge of PMDs state in cancer.

**A harmonized catalog of DMRs across 14 TCGA tumor types.** We applied Rocker-meth to a wide set of 5623 cancer samples across 14 TCGA tumor types profiled with HM450 array, comparing tumor samples with normal matching tissues (712 adjacent tissue samples, requiring >=10 normal samples per tumor type). First, upon selection of common tumor types, we compared the methylation values at overlapping DMRs obtained in the TCGA WGBS data with the larger array-based set and observed high concordance in terms of their delta beta values for almost all tumor types (Fig. 3a), despite the remarkably different sample sizes. LUAD discrepant behavior was likely due to lack in the WGBS dataset of CIMP-High samples, a common highly methylated subtype found in the original TCGA-LUAD description[45]. Of note, despite the different characteristics of data (i.e., array versus WGBS), we observed comparable DMR burden for hypo-methylated events in all tumor types, indicating that large demethylation events (hypo-blocks) detected by Rocker-meth are not affected by the different characteristics of the platforms (Supplementary Fig. 10a). For hyper-DMRs we observed less correlated signal but still comparable order of magnitude of differentially methylated genome estimated using the two strategies, probably due to the smaller fraction of hypermethylated genome than the hypomethylated counterpart, making it more prone to fluctuations dictated by a different platform or different sample size (Supplementary Fig. 10b).

Altogether, Rocker-meth identified a total of 16,423 hypo and 16,761 hyper DMRs in 14 tumor types dataset, with a median of 1352 hyper and of 1171 hypo DMRs across the datasets (Supplementary Fig. 11). DMRs have a range of abundance per dataset with extreme values for LIHC (largest number of DMRs) and THCA (no DMRs; dataset excluded from downstream analyses). The latter resulted in similar findings by applying Bumphunter to the same datasets, where the number of DMRs in THCA ($n = 148$) corresponded to only 6% of the median of the number of DMRs detected in all the other datasets ($n = 2908$) and include 95% (141/148) of DMRs that were supported by less than 6 sites (default threshold for DMR catalog inclusion). This data is in line with the TCGA-THCA study confirming that most samples have a methylation profile highly resembling the matching normal tissue[46].

Focusing on the 13 tumor types, we estimated that a median of 12.2% of cancer genome is characterized by DMRs (11.5% and 0.6% for hypo and hyper DMRs, respectively); the highest DMR burden (i.e, the fraction of cancer genome within DMRs) was observed for LIHC and BLCA, where DMRs span more than 30% of the genome (Supplementary Fig. 12). All tumor types but LIHC showed approximately concordant levels of hyper DMR burden, with values ranging from 0.1 to 1.2% (median = 0.62%)[46,47]. The full catalog of DMRs is reported in Supplementary Data 5. Altogether, these analyses also demonstrated the reliability of the Rocker-meth's algorithm and the robustness of the reported cancer-specific DMR catalog, enabling the inference of genome-wide methylation states, including large demethylation, from techniques with limited CpG coverage such as HM450.

**Inter and intra tumor type characterization of the TCGA DMR catalog.** In order to study the inter- and intra-tumor type DMR status heterogeneity, we performed a pan-cancer comparative analysis. The 13 tumor specific DMR sets were collapsed into one comprehensive set, that was further refined with a strict quality control procedure (see Methods and Supplementary Data 6). Interestingly, we found a substantial fraction of DMRs shared in more than half of the tumor types analyzed, with the highest recurrence observed for hypo-blocks (42%), followed by hyper (30%) and hypo (24%) DMRs (Fig. 3b). Conversely, DMSs reach a lower level of prevalence: we observed that about 10% of DMSs (AUC < 0.2 for hypo-methylation or AUC > 0.8 for hyper-methylation, lenient threshold) were shared in more than half of the tumor types, while this value drops to 2% when a more stringent threshold was used to call differential methylation (AUC < 0.1 or AUC > 0.9, stringent threshold). Interestingly, we estimated that tumor type-specific events correspond to about 20% for both hypo and hyper DMRs and 10% for hypo-blocks. On the other hand, tumor type-specific signal for DMSs was markedly higher, reaching up to 57% for stringent hyper-DMS events. Principal component analysis based on the integrated DMR atlas showed that the two first two components were associated with the Per Sample Fraction of Supporting Events (PSFSE) (i.e., burden of positive and negative differential methylation; Supplementary Fig. 13; see Methods). After removing the first two components, we applied UMAP analysis[47] and observed clear segregation of samples based on the difference of the beta values with respect to matched normal tissue (Fig. 3c). Of note, at the tumor type level, we observed cluster proximity depending on the organ of origin (KIRP, KIRC, and LUAD, LUSC) and female hormone-dependent tumors (BRCA, UCEC). Interestingly, we verified that the two distant clusters for ESCA reflected the two major subtypes[48] (Supplementary Fig. 14), one proximal to gastrointestinal cancers, and the other similar to squamous epithelial cancers. Similar observations on subtypes are also reported for BRCA and KIRP (Supplementary Fig. 15). Taken together, this confirms a clear influence of the cell of origin in shaping the DNA methylation changes that characterize tumorigenesis, in line with a previous report[38]. Interestingly, we observed that the most recurrent DMRs in each tumor type were also those showing on average higher prevalence across tumor types (Supplementary Fig. 16).

**Sample specific support of DMRs informs on patient specific alterations.** Rocker-meth's catalog refers to tumor type-specific DMRs. To estimate the extent to which each sample supported tumor type-specific DMRs, we implemented a strategy based on Z-score statistics that allows Rocker-meth to perform a single sample-wise analysis (Rocker-meth sample score) and applied this module to each of the TCGA cancer samples considered in this

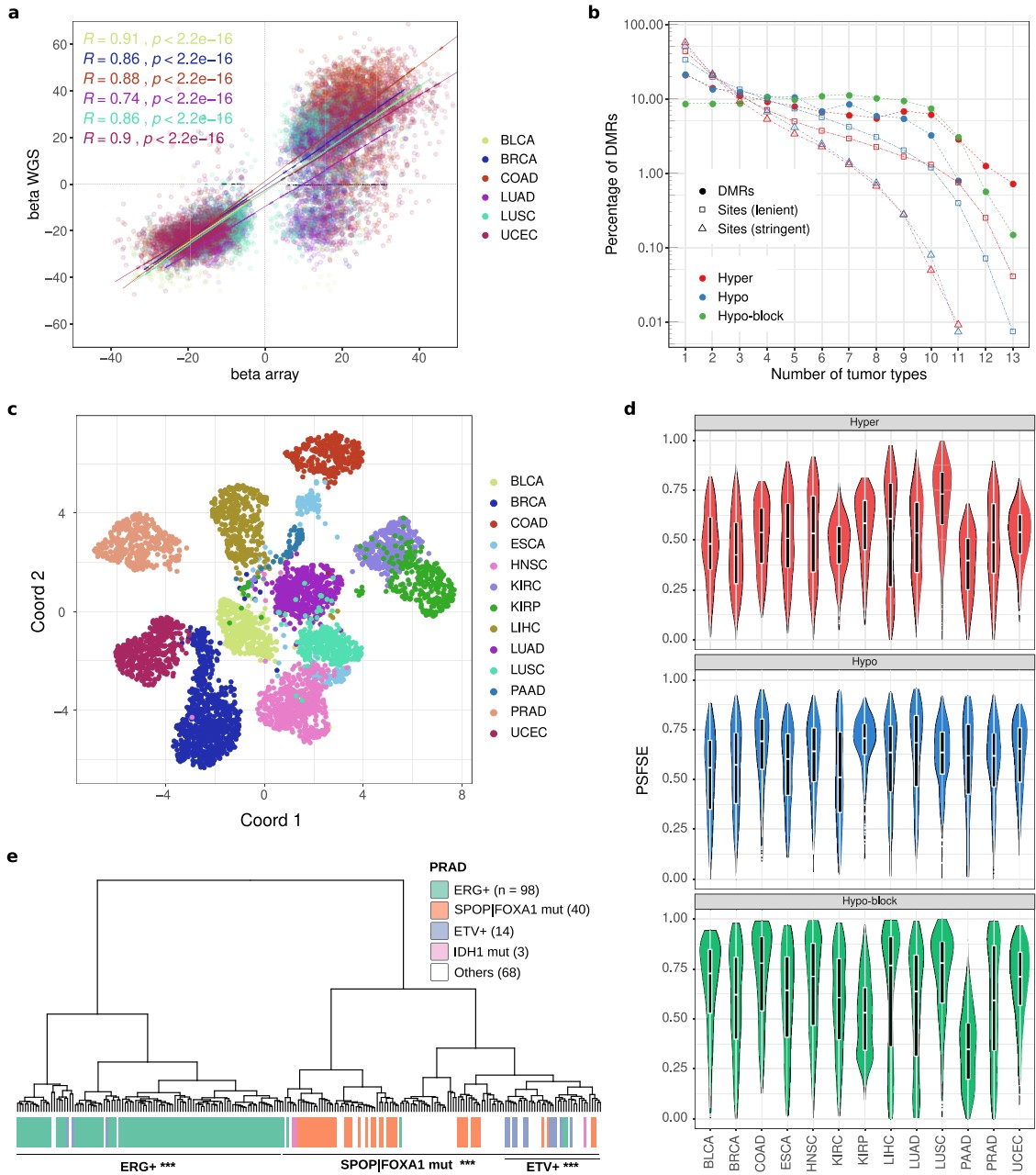

**Fig. 3 A Catalog of DMRs across multiple tumor types. a** Correlation between average beta difference in DMRs detected from WGBS data (y axis) and HM450 data (x axis). Pearson's correlation coefficients (R) and corresponding p-values are reported. **b** Fraction of shared differential methylation events across tumor types. Solid line: DMRs, dashed line: single CpG sites. Hypo DMSs and Hyper DMSs were selected using AUC below 0.2 and above 0.8 (lenient) or below 0.1 and above 0.9 (stringent). **c** Uniform manifold approximation (UMAP) of TCGA samples based on the average beta difference in each DMRs of the consensus set with respect to matched normal tissue. **d** Distribution of the per sample fraction of supporting events (PSFSE) for each class of DMRs (hyper: top, hypo: middle, hypo-block: bottom) across tumor types. **e** Unsupervised hierarchical clustering of TCGA-PRAD dataset based on DMRs z-scores. Molecular subtypes of tumor samples are reported.

study. Figure 3d reports the distribution of PSFSE across the 13 tumor types, using the cancer-specific catalog. Both for hyper and hypo DMRs, we observed that the majority (median 63%) of differential methylation signal was shared among all samples of a specific tumor type, with extreme values for some datasets such as 0 (PRAD) and 95% (LUAD and KIRC) for hyper-methylation events and 0.07% (LIHC) and >99% (LUSC) for hypo-methylation events.

Next, we performed unsupervised analysis via hierarchical clustering of the median beta values of DMRs calculated at the sample level. We focused on two datasets characterized in TCGA

genomic landscape studies by the presence of molecular subtypes with distinct DNA methylation patterns, PRAD and BRCA. Notably, in both cases, the DMRs detected by Rocker-meth were informative to segregate samples based on their molecular subtypes. In the prostate adenocarcinoma dataset (Fig. 3e) two main clusters were identified, one enriched for samples harboring mutations in SPOP or FOXA1 (SPOP|FOXA1mut) (hypergeometric test p-value < 2.2e-16, recall = 100%) and one enriched for samples characterized by the fusion transcript TMPRSS2-ERG (ERG+) (p-value < 2.2e-16, recall = 96%). Within the SPOP|FOXA1mut cluster, we observed a sub-cluster enriched

for samples characterized by fusion transcripts involving ETV (ETV+) ($p$-value $< 10^{-7}$, recall = 79%). In the breast cancer dataset, the tumor type-specific DMRs detected by Rocker-meth were informative to partially segregate PAM50 molecular subtypes, but the Normal-like one (Supplementary Fig. 17). In particular, we observed three main clusters characterized by the enrichment of luminal A ($p$-value $< 10^{-7}$, recall = 68%), luminal B ($p$-value $< 10^{-3}$, recall = 43%) and Basal ($p$-value $< 2.2$e-16, recall = 81%) samples. We also obtained a sub-cluster enriched for Her2 samples ($p$-value $< 10^{-7}$, recall = 43%). Supplementary Fig. 18 exemplifies the Rocker-meth sample scores for one DMR on 22q13.32 that is hypo-methylated in SPOP|FOXA1mut PRAD samples and barely ever in ERG+ samples. Further, Rocker-meth sample score indicates methylation event clonality at the locus that is significantly higher than in the SPOP|FOXA1mut group (median score $< -30$) than in all the other subtypes.

As we observed a substantial overlap between hypo-blocks and PMD, we studied the possibility to exploit sample-wise hypo-blocks burden (hypo-blocks PSFSE) as a proxy to the PMD-HMD score, recently developed to estimate cancer hypomethylation and found associated to mitotic cell division in cancer[40]. We found significant concordance between hypo-blocks burden and PMD-HMD score (Supplementary Fig. 19, $R = -0.62$, $p$-value $< 2.2$e-16). Altogether, these analyses demonstrate that the sample-based module of Rocker-meth also provides information indicative of patient-specific tumor features that is not redundant with the genomics.

**Gene wise and chromatin-wise analyses reveal distinct features of each DMR class**. We sought to characterize the functional role of hyper and hypo DMRs. First, we observed that the distributions of hyper and hypo DMRs around the closest Transcription Starting Sites (TSS) were markedly different (Fig. 4a). Considering DMRs within 10Kb from a TSS, both distributions showed a clear maximum in the proximity of the TSS; hyper-DMRs shows a relevant fraction of events (52%) mapping within 1Kb around TSS, while 29% of hypo-DMRs mapped around the TSS. Conversely, hypo-blocks were more broadly distributed around TSS with only 2% of events within ± 1Kb of the TSS, in line with their mostly intergenic localization. We then studied the genic localization of DMRs. Figure 4b reports the genomic annotation of hyper and hypo DMRs across tumor types (see Supplementary Data 7 for details). As expected, all the genomic features were differently represented in the three classes of DMRs: hyper and hypo DMRs were characterized by a prevalence of promoter-TSS and exon; 5′ UTR was markedly represented in hyper DMRs, while Transcription Termination Site and 3′UTR ($p$-value $< 3 \times 10^{-2}$) were more represented in hypo-DMRs. For hypo-blocks, we found a prevalence of intron and intergenic regions. As repetitive DNA sequences and transposable elements (Repetitive Elements, REs) are often aberrantly methylated in human cancers[49,50], we investigated the enrichment of REs mapping to the three classes of DMRs across all tumor types. Overall, we found a markedly different representation of RE classes within the classes of DMRs (Supplementary Fig. 20). Transposon associated elements such as Long Interspersed Nuclear elements (LINE) and Long Terminal Repeats (LTR) were more represented in hypo-blocks; Low Complexity and Simple Repeats elements that include CpG islands were prevalent in hyper-DMRs. A relevant fraction of hypo-DMRs maps to Short interspersed nuclear elements (SINEs). To further investigate the differential functional role of DMRs, we mapped DMRs to the 15 chromatin states defined by the ENCODE consortium in available matched normal tissues from the ROADMAP consortium[4,42] (see method section). We observed a similar representation of chromatin states across the different tumor types for the three classes of

DMRs (Supplementary Fig. 21). A selection of representative chromatin states and their distribution across the three DMR classes is reported in Fig. 4c (the distribution for all states is reported in Supplementary Fig. 22). Hyper and hypo DMRs show significantly higher overlap with Active and Flanking Active Transcription Starting Sites (TssA, TssAFlnk) compared to hypo-blocks. Hypo-DMRs show distinctive over-representation of Enhancers (Enh) compared to hyper-DMRs, while hyper-DMRs show specific enrichment of Bivalent Enhancers (EnhBiv) and Repressed Polycomb (ReprPC) states, in agreement with previous reports[51,52]. Furthermore, a region-based analysis using LOLA[53] highlighted a conserved enrichment of the cistrome of EZH2—the catalytic subunit of Polycomb Repressive Complex 2 (PRC2) that operates H3K27me3 deposition—from embryonic stem cells in hyper DMRs (Supplementary Fig. 23). In addition, we found that shared hyper-DMRs confirmed their evident co-localization with the EZH2 cistrome (Fisher Exact Test $p$-value $< 2.2$e-16, OR = 7.0)[52]. Finally, hypo-blocks displayed specific over-representation of inactive elements, including Heterochromatin (Het) and Quiescent (Quies) states, supporting the choice of using 10 Kbp to discriminate between functionally distinct events.

**Integrative analysis of DMRs and gene expression**. To study the role of DMRs associated with altered transcriptional programs, we performed an integrative pan-cancer analysis of DMRs detected by Rocker-meth and gene expression data from TCGA. Given that the function of DNA methylation depends on the genomic context[5,54,55], we decided to analyze the association between DMRs and differential expressed genes segregating DMRs based on their genic annotation. Figure 4d and Supplementary Data 8 reports the results of the association analysis between DMRs and differentially expressed genes across the 13 tumor types (full analysis including hypo-blocks is reported in Supplementary Fig. 24). As expected, we were able to confirm a strong enrichment of under-expressed genes for hyper-DMRs affecting their promoter/TSS regions (Fisher Exact Test, OR = 3.6, $p$-value $< 2.2$e-16, statistically significant in 11 out 13 of tumor types). Further, we found strong associations for hyper-DMRs in 5′ UTR (OR = 4.4, $p$-value $< 2.2$e-16, 11/13), and intron (OR = 1.2, $p$-value $< 2.2$e-16, 11/13) regions. For over-expressed genes, less marked yet significant associations were found. Top associations were found for hypo-DMRs in promoter-TSS (OR = 2.7, $p$-value $< 2.2$e-16, 9/13) and Intron (OR = 2.1, $p$-value $< 2.2$e-16, 6/13). Notably, we observed expected negative associations for hyper-DMRs in strong regulatory regions, such as promoter-TSS (OR = 0.83) and 5′ UTR (OR = 0.68). We then studied the associations observed between differential expression of a gene and the presence of concordant DMR (i.e., hyper for under and hypo for over-expression) affecting its regulatory genic regions. For this and downstream integrative DNA-methylation/gene expression analyses PAAD was excluded due to the low number of differentially expressed genes ($n = 308$), probably due to the low purity of tumor samples (median 38% by PAMES[56]). We observed that hyper-DMRs directly explained a median of 4.1% of all under-expressed genes, while hypo-DMRs explained a median of 0.39% of all over-expressed genes. The highest fractions of hyper-methylated and under-expressed genes were observed in COAD (7.3%), ESCA (5.6%), and PRAD (5.2%) (Supplementary Fig. 25). For all tumor types, hyper-DMRs in promoter/TSS contributed the most to the fraction of hyper-methylated and under-expressed genes with median values of 1.6%. For hypo-DMRs, introns contributed the most to the fraction of hypo-methylated and over-expressed genes with median values of 0.13%. Interestingly, comparable fractions were observed also for hyper-methylated and over-expressed genes

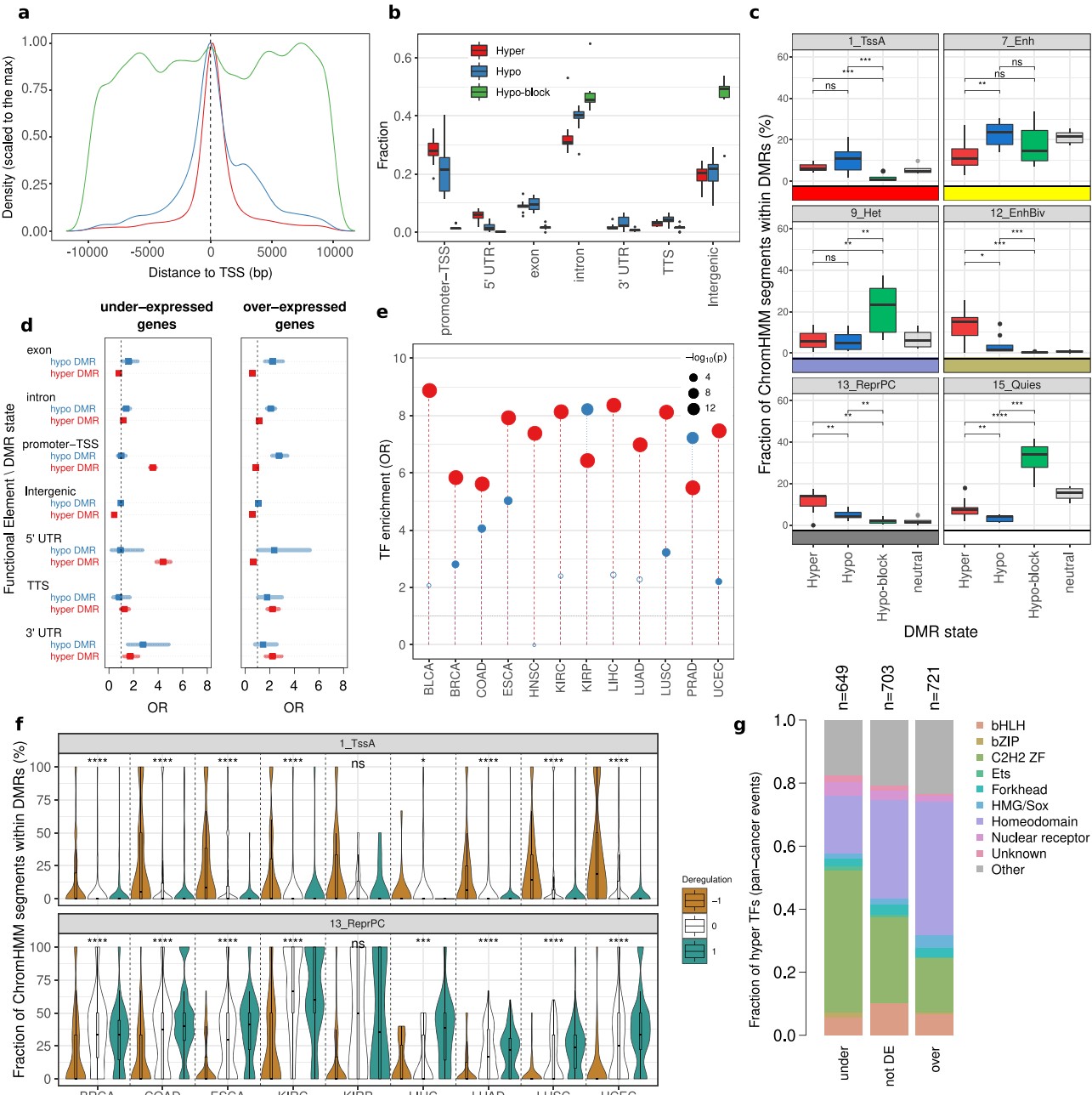

**Fig. 4 Functional and regulatory role of DMRs. a** Density plot of hyper (red), hypo (blue), and hypo-blocks (green) Differentially Methylated Regions (DMRs) around ±10 Kbp of TSS. **b** Box plot showing the distribution of the fraction of DMRs across the distinct tumor types annotated to the different genic features for hyper-DMRs, hypo-DMRs, and hypo-blocks. **c** Box plot of ChromHMM states from matched normal tissues. Each dot represents a tumor type, and the fraction of segments within each class of DMRs is reported. Neutral refers to the not-differentially methylated genome obtained from the Rocker-meth segmentation. Statistical significance is estimated using pairwise Wilcoxon test. **d** Dot plot showing the Odds-ratio of the pan-cancer enrichment of hyper (red) and hypo (blue) Differentially Methylated Regions (DMRs) in under-expressed (left) and over-expressed (right) genes estimated by Fisher Exact Test (FET) for different genic annotations. Error bars show the 95% confidence interval. **e** Lollipop plot reporting the Odds-ratio (OR) of the enrichment (by proportion test) of Transcription Factors in the set of differentially expressed genes deregulated by hyper-DMRs (red) and hypo-DMRs (blue) in their regulatory regions. The size of the dots associates (inversely) with p-values. **f** Fraction of matched ChromHMM states in DMRs associated with transcription factors, grouped by deregulation. -1: TF is downregulated, 0: TF is not deregulated. 1 TF is upregulated (FDR <0.05 is used to call differential expression). **g** Bar plots showing the distribution of TF families for the TFs affected by DMRs.

(median 1.9%, Supplementary Fig. 25b) and hypo-methylated and under-expressed genes (0.27%, Supplementary Fig. 25c). In summary, we observed that (i) alterations in DNA-methylation affect a small fraction of concordantly differentially expressed genes, and (ii) a not-negligible number of less characterized associations between DNA methylation and gene expression events exist.

**DMRs associated with transcription factors genes are affected by the underlying chromatin state.** We then performed specific analyses on these sets of differentially methylated and differentially expressed genes. First, we observed that for both hyper and hypo DMRs the distribution of the distance to TSS was different based on the direction of differential expression of corresponding genes (Supplementary Fig. 26). In particular, for hyper DMRs and

under-expressed genes we found that methylation essentially involved the region around TSS, while a more widespread methylation downstream to the TSS was observed for over-expressed genes. For both hyper and hypo DMRs, we found that first intron was predominantly affected compared to others (Supplementary Fig. 27), likely due to the enrichment of cis-regulatory elements in this genic feature[57]. Interestingly, TFs were significantly enriched in all of these sets of tumor-type specific hyper-methylated/differentially expressed genes (Fig. 4e, median OR = 7.5 by proportion test). For hypo-DMRs, we found significant over-representation of TFs for 7 out of 12 tumor types (median OR = 2.6). According to previous analysis, also for TFs we observed a marked fraction of discordant differentially methylated and expressed genes across all tumor types (Supplementary Fig. 28), in particular for hyper-methylated TFs. We therefore exploited chromHMM states to investigate the difference between under and over expressed hyper-methylated TFs.

Considering tumors' corresponding normal tissue ROADMAP data, we found that the hyper-DMRs involving under- and over-expressed TFs were characterized by different patterns of chromatin states (Fig. 4f, all chromatin states displayed in Supplementary Fig. 29). DMRs associated with under-expressed TFs showed enrichment of Active TSS (TssA) compared to over-expressed TFs, indicating that these genes were originally active in matched normal tissues, and likely present an over methylation that either precede or follows gene downregulation in tumors. On the other hand, DMRs associated with over-expressed TFs were characterized by the absence of TssA and by the presence of Repressed Polycomb (ReprPC) in matched normal tissue, indicating that these genes were originally silenced by PCR2 and thus low in DNA methylation. This specific pattern of chromatin states for activated/repressed genes through hyper-methylation was found significant for all tumor types with available normal tissue data, but KIRP. We then exploited a recent curated annotation of TFs[58] to investigate the difference between these two sets of hyper-methylated TFs. Interestingly, we found significantly different representation of TF classes between under-expressed and over-expressed hyper-methylated TFs (Fig. 4g and Supplementary Fig. 30). We observed that under-expressed TFs were largely C2H2 ZF, while over-expressed TFs were enriched for the class Homeodomain. These data were statistically validated in 9 and 7 out of 12 tumor types for under- and over-expressed hyper-methylated TFs, respectively (Supplementary Fig. 31), suggesting a common underlying process. Altogether, those results suggest a peculiar association between cancer DNA hypermethylation and gene deregulation depending on the baseline chromatin state (normal cells) in the proximity of key regulatory TFs.

**Application of Rocker-meth to single-cell DNA-methylation dataset from Colorectal Cancer patients**. Finally, to challenge the extent of the DMR catalog utility, we applied it to the single cell DNA-methylation (scMeth) data of 1265 cells from 10 CRC patients[41]. Upon assessment of the concordance of average beta difference values (primary tumor vs control) between TCGA-COAD and bulk-wise scMeth data ($R = 0.54$ for hyper, $R = 0.51$ for hypo DMRs and hypo-blocks, Supplementary Fig. 32), we applied the Rocker-meth's catalog to the scMeth data of 93 normal (NC) and 581 primary tumor (TP) cells and observed the expected segregation of cells based on their disease status (Fig. 5a). For TP cells, we found patient-specific segregation suggesting marked inter-tumoral heterogeneity. This observation was confirmed using the top 10% most variable 1Kbp tiling windows (Supplementary Fig. 33). Notably, we observed expected patterns of beta values in TP vs NC cells for all classes of DMRs. Following the same strategy applied on single samples from the

array-based TCGA dataset, we calculated the Per Cell Fraction of Supporting Events (PCFSE) and evaluated corresponding distributions in the scMeth dataset (Fig. 5b). Interestingly, we observed a patient-specific prevalence of hyper-DMRs, with CRC11 and CRC01 patients showing the lower fractions (PCFSE < 0.25) of supported events. However, for 9/10 patients we found a limited range for PCFSE, suggesting that most of the primary tumor cells from a given patient tend to support the same subset of events, thus indicating a low degree of intra-tumoral heterogeneity in most of DMRs. Patient-specific signal and low degree of intra-tumoral heterogeneity were also observed for hypo-DMRs. Of the three classes of DMRs, hypo-blocks show the higher values of PCFSE suggesting a low degree of intra- and inter-tumoral heterogeneity for this methylation class and their high prevalence across CRC cells, in line with our previous findings and a recent description of structural DNA methylation loss[59]. Interestingly, for hypo-DMRs and hypo-blocks in CRC01, we observed multimodal distribution for PCFSE, suggesting the presence of multiple methylation clones. This result was compatible with findings from the original study[41], where the authors found the presence of two lineages characterized by different genomic alterations and levels of genome-wide hypo-methylation. Indeed, for all DMR classes we observed different distributions of PCFSE between the two lineages, suggesting the utility of this per-cell measure in capturing heterogeneity in scMeth assays (Fig. 5c). Finally, we investigated the reliability of Rocker-meth's catalog in providing more detailed information about the clonal architecture of CRC01 tumor sample. Notably, UMAP analysis applied to beta values was able to segregate CRC01 primary tumor cells based on their sub-lineages, originally defined based on different sub-clonal SCNAs (Fig. 5d and Supplementary Fig. 34). Of note, the catalog reported could aid the interpretation of single cell DNA methylation data, mitigating the sparsity of measurements and allowing for comparisons with large-scale studies based on other technologies.

## Discussion
Rocker-meth is a computational tool to perform differential analysis of DNA methylation data, focused on the detection of DMRs and amenable to multiple experimental platforms data. The novelty of the method is the use of an ad hoc heterogeneous HMM algorithm to segment AUC values, which, as opposed to delta beta, are not dependent on the magnitude of the DNA methylation signal provided suitable sample sizes (Supplementary Fig. 35a). Specifically, in the context of small delta beta we observed that using AUC allows for a refined definition of DMR boundaries (Supplementary Fig. 35b). In principle, Rocker-meth enables the detection of both focal and large alterations, exploiting the sparseness of DNA methylation sites throughout the genome, without imposing artificial boundaries to the DMRs. Through a comprehensive parametric study and a wide range of test on synthetic and real data we demonstrated the capability of Rocker-meth to recapitulate known cancer features and define meaningful segments, obtaining excellent performance when compared with state-of-the-art tools. In particular, the comparative analysis on real data exploiting well-known association between DNA methylation and gene expression suggested that Rocker-meth may provide more functionally meaningful results compared to available state-of-the-art methods (Fig. 1d and Supplementary Fig. 8).

Importantly, this comparative analysis should be interpreted in the light of the lack of universal definition of DMR and on the absence of proper gold standards in terms of genome-wide methylation states. It is therefore remarkably difficult to properly compare the performance of different tools as often developed with different intents. This not only applies in terms of the

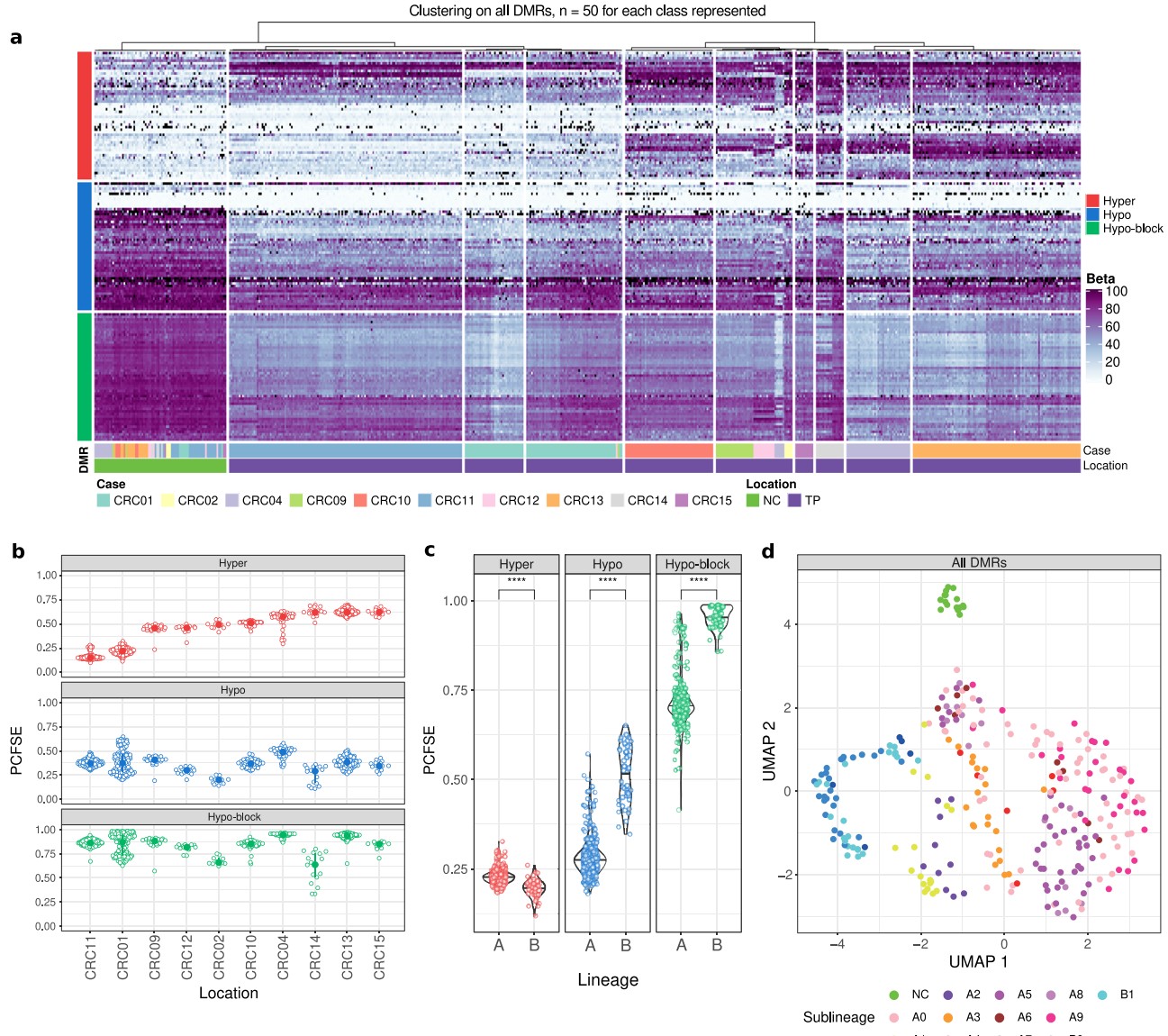

**Fig. 5 Application of DMR catalog to single cell DNA methylation from colorectal cancer samples. a** Heatmap of DMRs beta values in scTRIO cohort. 50 randomly selected regions for each class are depicted. For the dendrogram, hierarchical clustering using 1 - Pearson's correlation was applied to all cells having less than 10% of DMRs with missing values (NC: normal colon, TP: tumor primary). **b** Per Cell Fraction of Supporting Events (PCFSE) for each DMR class among the different patients. Only cells that had less than 50% of missing values within DMRs across each one of the DMR classes were considered. Patients are sorted based on PCFSE of Hyper DMRs. **c** Difference in PCFSE in lineage A and lineage B in CRC01 cells for each class of DMRs. Significance is evaluated using two tailed Wilcoxon test. **d** UMAP dimensionality reduction for cells from CRC01, using the beta value in all DMRs as input (missing values were imputed using KNN).

genomic size and magnitude of differential signal of potential DMRs, but also on the establishment of the required biological effect for a DMR to be functionally meaningful. For instance, very short differential signal at/around TFBS signal could be molecularly meaningful[8,60] but hard to distinguish from noise, therefore requiring orthogonal detection approaches and ultimately experimental validation. Relevant to certain applications, Rocker-meth does not implement covariate analysis, otherwise considered in methodologies such as DMRcate, DSS, and Bumphunter; similarly, it is not designed for the analysis of Differentially Variable Regions, offered by other tools[30,61,62]. All together these considerations elicit the selective use of either one approach based on the biological context or the parallel application of multiple complementary strategies.

By applying Rocker-meth to 6 WGBS cancer datasets, we observed that a large fraction of the cancer genome is involved in DNA methylation alterations with high level of consistency across tumor types. Specifically, we estimated that 31.4 and 0.71% of the cancer genome is characterized by gain and loss of methylation. When analyzing the characteristics of regions with gain and loss of methylation, we observed marked differences in terms of structure and genomic locations, including focal gain of methylation and widespread loss of methylation across tumor types. In particular, we found that loss of methylation was generally wider when affecting intergenic compared to genic regions. We therefore proposed a operative classification based on region length for loss of methylation regions, terming hypo-blocks as DMRs that display loss of methylation regions involving more than 10 Kb.

We observed that hypo-blocks significantly overlap with a recent curated catalog of tissue agnostic PMDs, indicating the potential of Rocker-meth to identify PMDs in unsupervised and tissue-specific manner.

While studies have presented pan-cancer characterization of DNA methylation based on single CpG sites[38,63–67], this is the first harmonized pan-cancer catalog based on DMRs built on more than 6000 human samples across all TCGA tumor types with available adjacent normal tissue data for differential analysis ($n \geq 10$). We observed consistent findings between WGBS and array-based data analysis in terms of DMR burden, length, and beta difference. In line with recent studies[68,69], we observed that different tumor types show similar DMR patterns and that a consistent fraction of DMRs is shared in more than half of the tested tumor types. Interestingly, we found that hypo-blocks constituted the class of DMRs with the highest prevalence, followed by hyper and hypo DMRs. Upon testing our method in the presence of a small set of controls, we found that Rocker-meth was able to well recapitulate original analysis (Supplementary Note and Supplementary Figs. 36–39); while we showed consistent results with imbalanced sets, we recognize that this depends on the homogeneity of the control set signal and that alternative shrinkage-estimator-based methods could be more suitable in case of marked heterogeneity or small sample size. Further, the catalog is defined under the assumption that the adjacent tissues represent proper controls, thus well capturing the cell-of-origin of corresponding tumors.

We then demonstrated the reliability of Rocker-meth to perform sample-wise analysis thanks to the intrinsic characteristic of DMRs to be extremely recurrent within a given tumor type, in line with our previous observation for methylation sites[56] and in contrast to genomic alterations such as Single Nucleotide Variants (SNVs) and Somatic Copy Number Variants (SCNAs). DMRs detected by Rocker-meth were informative in distinguishing the different tumor types and relevant subtypes. Relevant to clinical applications and precision oncology, Rocker-meth enables the assessment of DMRs on single sample data while identifying subtype-specific methylation events and clonality estimates through Z-score statistics. Notably, we have already applied successfully the Rocker-meth method to analyze plasma cfDNA methylation dynamics of castration-resistant prostate cancer patients[18].

As expected, the analysis of hypo-blocks revealed enrichment for intergenic regions, such as satellite repeats and transposable elements, while hyper and hypo DMRs preferentially map to regulatory genic regions, mainly within ±1Kb of TSS. Chromatin state annotation analysis clearly showed distinct functional roles between the three classes of DMRs and in particular between hypo-DMRs (regulatory) and hypo-blocks (global), supporting the value of our three-class DMRs classification.

Pan-cancer integrative analysis of DMRs and gene expression recapitulated well-known associations, such as the relationship between hyper-methylation in regulatory genic regions and under-expression[5]. For instance, hyper DMRs on the 5′UTR, promoter or TSS of a gene associate with gene under-expression. Similar, but less marked associations were found between hypo-DMRs and over-expression. Altogether though, about 4% of under-expressed and 0.4% of over-expressed genes can be linked to the presence of hyper and hypo DMRs in their regulatory regions (primary regulation). Interestingly, non *cis* regulatory effects of differential methylation in cancer have been recently reported, with relevant links to the response to immunotherapy[70] and metastatic potential[71]. While these associations were not explored in this work, our PSFSE represents a valuable metric to measure cancer specific DMR burdens and might be considered in such studies.

Next, we found significant enrichment of TFs[72] in differentially methylated and expressed genes. Overall, we observed similar representation of hyper-methylated under and over expressed TFs, suggesting that a relevant fraction of TFs can be regulated through context dependent effects. We found evidence that these two sets of TFs were characterized by different chromatin states. In particular, DMRs associated with over-expressed TFs were enriched for repressed polycomb state in the corresponding matched normal tissue, in line with the majority of these genes being members of the homeodomain class of TFs. These results are consistent with a recent report that demonstrated the interplay between DNA hypermethylation in cancer and de-repression in a subset of genes[73].

Finally, we demonstrated the utility of our catalog of DMRs for the analysis of scMeth data, characterized by sparse and noisy signal. Our analysis recapitulated expected results based on original findings and confirmed the reliability of DMRs inferred from TCGA data[41]. Distinguishing cells based on their lineage and sub-lineage membership, we also demonstrated that Rocker-meth's catalog was able to capture intra-tumor heterogeneity.

We anticipate that our method could be applied in other contexts where large numbers of control and test samples are available. The comprehensive robust catalogs of DMRs here provided represents a novel resource in two main settings. First, the characterization of DMRs allows for the integration of other molecular or phenotypic layers in a functional genomics context, providing a detailed map of DNA methylation deregulation in primary tumors. Second, future studies focusing on the development and identification of non-invasive cancer biomarkers, such as those describing new DNA-methylation-based assays for the analysis of cell-free tumor DNA, will be able to exploit the signal from the catalog of DMRs, thus augmenting statistical power for small cohorts. We envision that cancer-specific and pan-cancer DMRs, either provided here or generated by applying Rocker-meth to other datasets of interest, could facilitate the development of multi-purpose liquid biopsy tests for the early detection of cancer and the monitoring of patients' treatment response.

## Methods

**The Rocker-meth tool**. Rocker-meth (Receiver operating characteristic curve analyzer of DNA methylation data, Fig. 1) consists of four main modules: (1) computation of Area Under the Curve (AUC) values from Receiver operating characteristic (ROC) Curve analysis of methylation levels (i.e., beta values) in tumor versus normal samples; (2) segmentation of AUC values by a tailored heterogeneous Hidden Markov Model (HMM); (3) estimation of intra-segment homogeneity by Wilcoxon–Mann–Whitney (WMW) test on beta values of CpG sites in tumor versus normal samples; (4) identification of sample-specific DMRs by Z-score statistics. These steps are described in the following text.

*Computation of AUC values (step 1)*. DNA methylation status is usually reported as the fraction of alleles that are methylated (beta value, ranging from 0 (unmethylated) to 1 (fully methylated)). To identify Differentially Methylated Sites (DMSs), we computed the Area Under Curve (AUC) of a Receiver Operating Characteristic (ROC) curve for each site in each cancer type. ROC curves display the accuracy of a binary classification, which assumes hyper-methylation in tumor samples. Thus, AUC scores close to 1 identify optimal segregations between tumor and normal samples with tumor samples on average showing beta values greater than normal samples (hyper-methylation). On the contrary, AUC scores close to 0 correspond to sites in which tumor samples demonstrate on average lower beta values than normal samples (hypo-methylation).

*Segmentation of AUC values (step 2)*. In order to identify differentially methylated regions (DMRs), we exploited a strategy inspired by our previous work on the analysis of Copy Number Alterations and runs of homozygosity[39,74]. We modeled AUC values by means of a discrete state hidden Markov model (HMM) with continuous output. We model HMM states by Gaussian emission probabilities. Supplementary Fig. 40 reports normal qq-plot analysis of the AUC scores across the 14 TCGA datasets considered in this study. This analysis supports the use of such a model. A discrete HMM with continuous output is characterized by the following elements:

- The number of hidden states, $K$, in the model. The states are denoted as $S = \{S_1,...,S_K\}$ while $q_i$ denotes the actual state at position $i$ ($1 \le i \le n$).
- The observed data $O = \{O_1,...,O_N\}$.
- The initial state distribution, $\pi$, where $\pi_{1k} = P(q_1 = S_k)$.
- The emission probability distributions $b_k(i)$ that is the probability of observing $O_i$ at position $i$ given the state $S_k$: $b_k(i) = P[O_i \mid q_i = S_k]$
- The transition matrix, $A$, giving the probability of moving from one state to another, $A_{lm} = P(q_{i+1} = S_m \mid q_i = S_l)$ for $1 \le i \le n - 1$ and $1 \le l, m \le K$.

To model our problem, we used a three-state HMM ($K = 3$) where the hidden states represent hypo-methylated ($S_1$ = hypo-methylation), not differentially methylated ($S_2$ = no differential methylation) and hyper-methylated ($S_3$ = hyper-methylation) states of the genome and the observations are the $AUC$ values at each CpG position $i$ ($AUC_i$).

The emission distributions are truncated Gaussian densities with the following form:

- $P(AUC_i \mid q_i = S_1) = g_l^u(AUC_i; \theta_1)$
- $P(AUC_i \mid q_i = S_2) = g_l^u(AUC_i; \theta_2)$
- $P(AUC_i \mid q_i = S_3) = g_l^u(AUC_i; \theta_3)$

Where $\theta_1 = (\mu_1, (1-F) \cdot \sigma_{Tot})$, $\theta_2 = (\mu_2, F \cdot \sigma_{Tot})$ and $\theta_3 = (\mu_3, (1-F) \cdot \sigma_{Tot})$ are the means and the variances and $l$ and $u$ are the lower and upper bounds ($l = 0$, $u = 1$) of the three truncated gaussian densities $g$. $F$ is a parameter used to split the total AUC signal variance in two parts: the variance of not differentially methylated state and the variance of differentially methylated states. $\sigma_{Tot}$ is estimated by calculating the standard deviation of all AUC values.

Finally, to take into account the distance between consecutive CpG sites $d = (d_1, d_2,..., d_{n-1})$, we decided to incorporate them into the transition probabilities matrix $A_i$ defined for $1 \le i \le n - 1$:

$$A_i = \begin{pmatrix} 1 - p(1 - e^{-f_i}) & \frac{p(1-e^{-f_i})}{2} & \frac{p(1-e^{-f_i})}{2} \\ \frac{p(1-e^{-f_i})}{2} & 1 - p(1 - e^{-f_i}) & \frac{p(1-e^{-f_i})}{2} \\ \frac{p(1-e^{-f_i})}{2} & \frac{p(1-e^{-f_i})}{2} & 1 - p(1 - e^{-f_i}) \end{pmatrix},$$

where $p$ represents the probability of moving from one state to another in the homogeneous HMM, $f_i = d_i / d_{Norm}$ and $d_{Norm}$ is the distance normalization parameter. The parameter $d_{Norm}$ modulates the effect of genomic distance $d_i$ on the transition probabilities: the larger $d_{Norm}$ is, the smaller is the probability to jump from one state to another.

We used $\mu_1, \mu_3$ (the mean of hypo-methylation and hyper-methylation states), $p$ (transition probability) as algorithm parameters instead of estimating them with an Expectation-Maximization (EM) algorithm; the rationale is they well define the methylation states ($\mu_1$ and $\mu_3$) and set the resolution of our computational method (the capability to detect DMRs of different size, $p$). Finally, once all the parameters have been set, the Viterbi algorithm is exploited to find the best state sequence and consequently to associate each AUC value to one of the three states, thus identifying AUC segments.

*Intra-segment homogeneity (step 3).* For each AUC segment, intra-segment homogeneity is assessed through the Wilcoxon–Mann–Whitney test between the CpG-wise averaged beta values in cancer samples and normal samples. Resulting $p$-values are then adjusted by FDR (i.e., Benjamini–Hochberg procedure)..

*Rocker-meth Sample Score (step 4).* A tailored $Z$-score statistics strategy was developed to enable sample-wise analysis of DMRs. For each DMR $z$ and tumor sample $i$ Rocker-meth Sample Score (*RockerSS*) is calculated by the following formula:

$$RockerSS_{iz} = \frac{(\bar{\beta}_{iz}) - median_j(\bar{\beta}_{jz})}{mad_j(\bar{\beta}_{jz})},$$

where $j$ indicates the index over normal samples, $(\bar{\beta}_{iz})$ and $(\bar{\beta}_{jz})$ are the median beta values of CpG sites within a DMR $z$ for tumor and normal samples, respectively, and $median_j$ and $mad_j$ are the median and the maximum absolute deviation of the $(\bar{\beta}_{jz})$ across normal samples $j$. Expressing beta values as the percentage of methylation, to avoid over-estimation of RockerSS in cases for which $mad_j(\bar{\beta}_{jz}) < 1, \%$ (a single percentage point of DNA methylation), we set $mad_j(\bar{\beta}_{jz}) = 1$.

To control for the impact of the sparseness of an assay design on the DMR lengths (as in the case of certain arrays), the tool includes a parameter (*max_distance*) to allow users to intentionally split DMRs if adjacent CpG sites are more distant than a user-defined selectable value.

Rocker-meth was developed as a package for the statistical language R (https://cran.r-project.org/) and is freely available under MIT license at https://github.com/cgplab/Rocker-meth.

**Synthetic datasets.** We considered 15 synthetic DNA methylation datasets emulating Infinium Human Methylation 450 K BeadChip (HM450, $n = 5$), reduced bisulfite sequencing (RRBS, $n = 5$) and whole-genome bisulfite sequencing (WGBS, $n = 5$) data. In particular, we considered 8 datasets from the Metilene website (http://www.bioinf.uni-leipzig.de/Software/metilene/) (class 1 to 4, as in Jühling et al.[33]); 2 datasets were generated with the metilene package using the scripts simulate_DMRs_RRBS.R and simulate_DMRs_WGBS.R with Mixture Factor $c = 0.55$ to mimic more challenging signal-to-noise ratios (here referred to as class 5) (Supplementary Data 1). To generate the HM450 synthetic datasets, we modified the original simulate_DMR_RRBS.R script to simulate synthetic data at HM450 specific CpG sites (restricted to chromosome 10). The script is available at the zenodo repository https://doi.org/10.5281/zenodo.2586588 (simulate_DMR_450k.R). Given the non-uniform distribution of the HM450 probes along the genome (genic vs intergenic), to ensure appropriate coverage of DMRs we set the following parameters: min DMR length = 5, max DMR length = 10, number of DMRs = 20 for non-promoter regions; min DMR length = 8, max DMR length = 20, number of DMRs = 60 for promoter regions.

**Comparison study on synthetic data.** We applied Rocker-meth, Metilene version 0.2-4[33], DSS version 2.28.0[31], DMRseq version 1.0.13[32], DMRcate version 1.16.0[30], and Bumphunter version 1.22.0[29] on the 10 RRBS and WGBS synthetic datasets. For all methods, we used default parameters. For HM450 datasets, we considered only array DMR detection tools (i.e., Rocker-meth, Bumphunter, and DMRcate) and metilene. Rocker-meth was run with $p = 0.05$, $F = 0.4$, $D_{norm} = 10^5$, mu = 0.25 (default parameters). AUC segments with WMW $p$-value (FDR) $\ge 0.05$ or comprising less than 6 data points were discarded for downstream analysis. The evaluation of the performance of all algorithms was assessed by precision, recall, and F1 statistical measures as in Jühling et al.[33] and specificity. For segment-wise statistics, we required that predicted DMRs have an overlap greater than 30% with simulated DMRs. For the computational time, we considered the mean of the running times in RRBS and WGBS synthetic datasets.

**Cancer Datasets.** We downloaded DNA methylation data of 14 tumor types of The Cancer Genome Atlas from the GDC Legacy Archive (https://portal.gdc.cancer.gov/legacy-archive/search/f); specifically, BLCA (Bladder Urothelial Carcinoma), BRCA (Breast invasive carcinoma), COAD (Colon adenocarcinoma), ESCA (Esophageal carcinoma), HNSC (Head and Neck squamous cell carcinoma), KIRC (Kidney renal clear cell carcinoma), KIRP (Kidney renal papillary cell carcinoma), LIHC (Liver hepatocellular carcinoma), LUAD (Lung adenocarcinoma), LUSC (Lung squamous cell carcinoma), PAAD (Pancreatic adenocarcinoma), PRAD (Prostate adeno-carcinoma), THCA (Thyroid carcinoma), and UCEC (Uterine Corpus Endometrial Carcinoma). DNA methylation values were originally generated using Illumina HumanMethylation450 BeadChip (http://www.illumina.com). These datasets were selected for the presence of at least 10 normal adjacent tissue samples for each tumor types, in order to provide a sufficient number of cases in the control group. Upon exclusion of metastatic tumors and of duplicated experimental data, this dataset comprises a total of 5623 tumor samples and 712 normal samples. Gene expression data were downloaded as raw counts from Recount2 (https://jhubiostatistics.shinyapps.io/recount/). Upon exclusion of metastatic tumors, this dataset comprises a total of 6719 tumor samples and 643 normal samples. Details about the DNA methylation and gene expression datasets are reported in Supplementary Data 9–10. Beta tables and AUC are also available in a zenodo repository (https://doi.org/10.5281/zenodo.2586588). WGBS data from Zhou et al.[40] were downloaded from http://zwdzwd.io/trackHubs/TCGA_WGBS/hg19/bw_mindepth5/. For Fig. 3c–e we considered only cancer samples with available PAMES (cancer-specific thresholds) tumor purity estimates >0.5, as reported in Benelli et al.[56].

**Application of Rocker-meth and other state-of-the-art methods in TCGA datasets.** We applied Rocker-meth on 14 tumor types for a total of 5623 tumor samples and 712 normal samples using default parameters ($p = 0.05$, $F = 0.4$, mu = 0.25, and Dnorm = $10^5$). In particular, we applied Rocker-meth to each tumor type dataset considering the primary tumor samples (TP) as the test set and the corresponding normal tissue samples (NT) as the control set. Only DMRs with FDR <0.05 and supported by at least 6 sites were considered for downstream analysis. The threshold of the number of sites for nominating reliable DMRs was selected based on specific analysis on synthetic data, as reported in Supplementary Note. Bumphunter (version 1.22.0) and DMRcate (version 1.16.0) were also applied to the same data using default parameters.

**Genomic annotation and functional role of DMRs.** Association of DMRs to different genomic features, assignment of the closest transcript and computation of the distance to closest TSS were done by the annotatePeaks function of the HOMER package[75], using RefSeq as gene set and hg19 as genome build (Supplementary Data 11). To calculate the association between DMRs and repetitive elements (REs), we downloaded RepeatMasker data from UCSC table browser (https://genome.ucsc.edu/cgi-bin/hgTables, June 2017). BEDTools intersect[76] with –f 0.99 was used to associate DMR to REs. Inferred chromatin states for the hg19

genome were downloaded from (https://egg2.wustl.edu/roadmap/data/byFileType/chromhmmSegmentations/ChmmModels/coreMarks/jointModel/final/, Roadmap Epigenomics consortium). For each of the analyzed tumor types, we queried the Encode dataset for the matched tissue of origin. A total of 10 projects presented a suitable normal tissue profile (BRCA:E119 - HMEC Mammary Epithelial Primary Cells, LUAD:E096 - Lung, LUSC:E096 - Lung, PAAD:E098 - Pancreas, LIHC:E066 - Liver, ESCA:E079 - Esophagus, COAD:E101 - Rectal Mucosa Donor 29, KIRP:E086 - Fetal Kidney, KIRC:E086 - Fetal Kidney, UCEC:E097 - Ovary). For each tumor specific segmentation of methylation (including neutral segments), the fraction of chromHMM segments that are within segments were computed for the 15 chromatin states[4,42]. As a result, each chromHMM segment was assigned to a differentially methylated state if it was completely within a methylation segment. To calculate the co-localization between hyper-DMRs and EZH2 biding sites, we downloaded the ENCODE Regulation 'Txn Factor' track from UCSC database (http://hgdownload.cse.ucsc.edu/goldenpath/hg19/encodeDCC/wgEncodeRegTfbsClustered/) and considered only binding sites supported by more than half of the samples (cell types). Statistical significance of the overlap between hyper-DMRs and EZH2 binding sites was assessed by BEDTools fisher using –F 0.5.

**Integrative analysis of DMRs and gene expression.** Homogeneous transcriptomic data for the TCGA datasets were obtained from the recount2 resource[77]. After count rescaling using the recount R package, differential expression analysis was performed using DESeq2[78] comparing Primary Solid Tumor samples against Solid Tissue Normal samples for each tissue type. The pan-cancer analysis of differentially expressed genes is available at https://doi.org/10.5281/zenodo.2586588. To associate each DMR to the corresponding gene, the annotatePeaks function of the HOMER package[75] was used. For Supplementary Fig. 26, the distance between each DMR and TSS was calculated by the sum of the distance to TSS estimated by HOMER and half-length of the DMR. Transcripts lengths were downloaded from Ensembl Biomart (August 2017)[79]. For transcription factors, we used the manually curated list reported in ref. [58]. To nominate the state of DMRs associated with transcription factors we queried ENCODE tissue data for a subset of tumor type as previously described. In this case we overlapped each DMR with the chromHMM segments and obtained a fractional representation of the chromatin states within the same tissue in the matching normal samples.

**Comparison study of Rocker-meth, Bumphunter, and DMRcate in TCGA datasets.** To assess the performance of Rocker-meth, Bumphunter, and DMRcate on real data, we evaluated the ability of each method in recapitulating well-known features of DNA-methylation in human cancers by odds-ratios (OR). For the presence of hyper-DMRs in the promoter, TSS or 5′ region of a gene and its under-expression, we estimated ORs from the proportion of hyper-DMRs being in the promoter, TSS or 5′ region of a gene and the proportion of genes being under or over-expressed. For CpG islands, we estimated ORs from the proportion of DMRs mapping to CpG islands being hyper-methylated and the proportion of DMRs mapping to CpG islands being hypo-methylated. For intergenic regions, we estimated ORs by the ratio between the proportion of DMRs mapping to intergenic regions being hypo-methylated and the proportion of DMRs mapping to intergenic regions being hyper-methylated.

**Per sample/cell fraction of supporting events.** To estimate the recurrence of DNA methylation events, we used the following strategy. For each tumor type, we considered the hyper- and hypo- DMRs, indicating as $n_{hyper}$ and $n_{hypo}$ the total number of DMRs identified. We then considered the Rocker-meth sample score (RockerSS) of each DMR $z$ in tumor samples $i$. Per sample fraction of supporting events (PSFSE) were calculated as follows:

$$PSFSE_{i,hyper} = \frac{\sum_z \Theta(RockerSS_{zi} - 3)}{n_{hyper}},$$

$$PSFSE_{i,hypo} = \frac{\sum_z \Theta(-RockerSS_{zi} + 3)}{n_{hypo}},$$

where $\Theta$ is the Heaviside function. The selected threshold $(+/-3)$ is reasonably defined to minimize false positives.

**Analysis of TCGA WGBS dataset.** We applied Rocker-meth with default parameters to 6 tumor types (BLCA, BRCA, COAD, LUAD, LUSC, and UCEC) comprising 6 normal and 27 cancer samples[40]. Normal samples were pooled together and used as a reference for each tumor type due to the low sample size. Pancancer set of DMRs was retrieved applying Rocker-meth to the 27 cancer samples. DMRs from WGBS were then annotated using the annotatePeaks function of the HOMER package and overlapped with the DMRs from the Illumina HM450 data using the R package GenomicRanges[80].

**Pan-cancer catalog of DMRs.** Starting from significant DMRs obtained for each tumor type, 'bedtools --multiinter' clustering[76] was used to define shared and private regions of differential methylation across 13 tumor types. After this step,

further refinement of the integrated DMR set was performed applying the following strategy:

- For each tumor type, we compute the average Beta median of each integrated DMR in each normal and tumor sample. Consistently with previously applied filters, DMRs with less than 6 probes are discarded.
- Median Beta values are tested with Wilcoxon–Mann–Whitney test (two-sided) comparing normal tissues and primary tumors. After applying Benjamini–Hochberg correction, DMRs with FDR <0.05 and mean beta difference >5 were considered significant.
- In addition, only DMRs with at least 3 sites with concordant differential signals (AUC <0.25 for hypo, AUC >0.75 for hyper) were retained.
- Last, the threshold of 10 kb was used to distinguish short hypomethylated regions and hypo-blocks, following the criteria used for tumor-type specific catalogs.

Those additional checks were designed to guarantee the good quality of pan-cancer segments, given the possibility that collapsing the single DMR sets together could generate small regions that might not retain the expected signal. The resulting refined pan-cancer DMR catalog contains a total of 8098 regions (Supplementary Data 6). For the pan-cancer DMRs analyses of shared and private DMRs, segments that were detected in multiple projects but presented no dominant state across projects were discarded.

**Dimensionality reduction and clustering.** Starting from the previously defined integrative DMR atlas, for each tumor sample we computed the difference between each tumor sample DMR beta and the median beta in normal samples from the same cohort. DMRs in chromosome X and Y were excluded to avoid biases due to uneven sex distribution. Fast truncated principal component analysis was performed using the 'irlba' R package, setting the number of components to 50, center = TRUE and scale = FALSE. The first 30 components explained more than 95% of the total variance, with the first 2 accounting for nearly 50% of the variance. Following the observation that the first two principal components were mainly associated with hyper and hypo/hypo-block PSFSE (Supplementary Fig. 13), we excluded them and used the next 13 ($n$ = number of analyzed tumor types) components to perform a non-linear dimensionality reduction. UMAP analysis was performed using the 'uwot' R package with normalized laplacian initialization, euclidean distance and parameters $a = 2$, $b = 1.6$. For BRCA, ESCA, and KIRP subtypes, we queried a comprehensive annotation using the TCGAbiolinks package[81].

**PMD/HMD analysis.** Previous definition of partially methylated domains (PMD) and highly methylated domains (HMD) regions genome wide for hg19 genome was obtained from https://zwdzwd.github.io/pmd[40]. We utilized the stringent definition of PMD/HMD based on Beta standard deviation across samples. Of note, the utilized definition has been obtained using a bulk of different cancer types and normal samples. To define pairs of events between WGBS Rocker-meth segmentation and PMD/HMD regions a minimum overlap of 30% of the region was required.

**Analysis of scTrio DNA methylation dataset.** Single cell DNA methylation values from Bian et al.[41] were downloaded from the GEO repository GSE97693. HEp-2 cells were discarded. Genomic-based lineage, sub-lineage, and location information were obtained from the original work. For each DMR from the COAD catalog (TCGA based) we computed the mean Beta value for each cell, removing missing values. This resulted in a matrix with columns equal to the number of cells and rows equal to the number of DMRs. The overall Beta difference was measured for each DMR based on pooled normal colon cells (NC) and pooled tumor primary cells (PT) and compared with results from TCGA-COAD. For heatmap visualization, we filtered out cells with more than 10% of missing values across DMRs. Single cell z-scores were computed using the previously reported formula and using the pool of normal cells as control. For the z-score analysis in Fig. 5b, all cells that had more than 50% of missing values within DMRs for at least one class were discarded. Even though the aggregation by DMRs greatly reduced the inherent sparsity of data, a process of imputation was necessary to allow for further dimensionality reduction analysis. DMRs without Beta values were imputed using the 'impute.knn' function from the impute package. For the analysis focused on patient CRC01 cells, we added also all the cells from available metastatic lineages. Dimensionality reduction was performed using UMAP[47] on the first 50 principal components with parameters $a = 2$, $b = 1.6$, norm-laplacian initialization, and 1-Pearson correlation as distance metric.

**Data analysis and visualization.** Statistical analyses, data processing, and visualization were performed using the R environment (R Core Team, http://cran.r-project.org/) and the tidyverse package[82]. For box plots, lower and upper bars correspond to the minimum and maximum non-outlier values of the data distribution. Outliers are defined as values outside of the range (Q1 − 1.5 × (Q3 − Q1), Q3 + 1.5 × (Q3 − Q1)), where Q1 and Q3 are the first and third quartile, respectively.

**Reporting summary**. Further information on experimental design is available in the Nature Research Reporting Summary linked to this paper.

## Data availability

Source data are available in Supplementary Data 1–11. Additional data are available at https://doi.org/10.5281/zenodo.2586588.

## Code availability

Rocker-meth is freely available under MIT license at https://github.com/cgplab/Rockermeth.

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

## Acknowledgements

We thank the members of the Laboratory of computational and functional oncology (University of Trento) and the Bioinformatics unit of the Hospital of Prato for fruitful discussions. We thank the authors of scTrio2 seq paper, in particular Dr. Shuhui Bian, for the help interpreting scDNA methylation data from CRC patients. We are grateful to prof. Alessio Zippo for the insightful comments on the manuscript and to Giordano Alvari for the initial layout of Fig. 1a. The results shown here are in whole or part based upon data generated by the TCGA Research Network: https://www.cancer.gov/tcga. This work was supported by the Fondazione CR Firenze (to MB); Italian Minister of Health GR-2018-12365195 (to MB); the European Research Council (ERC) under the European Union's Horizon 2020 research and innovation programme (grant agreement no. 648670; SPICE) (to FD); NCI P50 CA211024-01A1 (to FD); Fondazione AIRC per la Ricerca sul Cancro, Accelerator Award 2018 ID 22792 (to FD).

## Author contributions

Conceptualization: M.B., G.M.F.; Methodology: M.B., A.M., G.M.F., D.R.; Software: A.M., D.R.; Data interpretation: M.B., G.M.F., D.R., C.B., I.M., L.M., F.D.; writing-review and editing: all authors; data interpretation: all authors; writing-original draft: M.B., G.M.F., F.D.; funding acquisition: M.B., F.D.; resources: D.R., G.M.F.; supervision: M.B., F.D. All authors read and approved the final manuscript.

## Competing interests

The authors declare no competing interests.
