## [Peer Review File · Communications Biology]

Reviewers' comments:

Reviewer #1 (Remarks to the Author):

In this manuscript, Benelli et al. develop a new method called Rocker-meth for identifying differentially methylated regions in cancer. Rocker-meth uses an AUC segmentation to detect region boundaries, and is particularly robust to noises. Furthermore, the method is applicable to both array and bisulfide sequencing platforms. The authors demonstrate the performance of Rocker-meth with both simulation datasets and TCGA datasets.

A big part of the work is a pan-cancer analysis of DNA methylation profiles. By dividing differentially methylated regions into hypo, hyper, and hypoway, they show cancer genomes harbor substantial hypomethylations, particularly in intergenic regions. In addition, they demonstrate the associations between DMRs and genetic features, and similarly the association between methylation and transcriptional readouts. An interesting observation is that many of the MDRs are shared across cancer types. Finally, they analyzed single-cell methylation data with the new tool and demonstrate the tool can work well for single-cell datasets.

Overall, the manuscript is well written. While the pan-cancer analysis is largely confirmatory, the results lend further validation in addition to the benchmark experiments. I think the paper is of high quality, and thus, endorse its publication. I have the following comments.

The authors show high correlation between array and sequencing-based results in Fig S10. Notably, they identified many more hypomethylation events with sequencing data than array data (31% vs 11.5% of the cancer genome). This analysis was not done for hypermethylation. Is it because 450k arrays do not have sufficient coverage in intergenic regions that caused this disparity? For whatever reason, it would be helpful to discuss and offer some insights.

The second issue is about the single sample analysis. Rocker-meth identifies DMR boundaries based on group comparisons (tumor vs normal), and assumes the boundary applies to each sample when evaluating single tumor samples. It is unclear, however, how heterogeneous the tumor samples are with regard to hypo or hyper-methylated regions. They could share a fraction of the segment but not entirely, in a similar scenario to copy number events. They could be particularly relevant if the targets of methylation events are positively selected during cancer evolution. It is thus important to assess how Rocker-meth performs in this aspect.

Reviewer #2 (Remarks to the Author):

In the manuscript "Cancer methylomes characterization enabled by Rocker-meth" by Benelli, Franceschini, et al. the authors present a computational method, Rocker-meth, for calling differentially methylated regions (DMRs). The authors then apply the approach to TCGA data and perform extensive data analyses to characterize the DMRs they called.

The manuscript has two relatively disjoint parts. One part being bioinformatics methods oriented introducing a new DMR calling procedure and providing a few comparisons with existing approaches. The other part was extensive biological focused analyses of the DMRs including with integration of various other genomic data where the specific choice of method to call DMRs seemed secondary. I felt the new bioinformatics method received more motivation from the abstract and introduction, but most of the results section and the lengthy supplement were focused on the biological analyses. Since the authors only used their method for the biological analyses, it was not clear that the author's new method actually enabled most of the biological results as suggested by the manuscript title. For these two parts to fit well together in the same

manuscript, I think the authors should make a stronger case that biological results they are presenting are actually enabled by their new DMR calling procedure relative to existing methods for the same problem. For which analyses, if any, would different biological conclusions have been reached if an existing DMR calling procedure was used and why are the authors confident they are correct in such cases?

Figure S8, is the only comparison on non-simulated data. The authors are comparing odds-ratio without considering coverage of the predictions. How does the coverage of the different methods compare in these evaluations? If the competing methods have greater coverage then it is difficult to compare the methods just based on the odds ratios. Also do the authors have an explanation for why both existing methods compared to are going in the opposite direction for the 'Under-expression/Hyper in promoter/TSS' category?

I am concerned about the potential sensitivity of the method to the settings of a number of arbitrary parameters and that the evaluations relative to other methods did not adequately consider that. In particular in the "Comparison study on synthetic data" in the methods section the authors state that the default parameters for all methods were used, which were the main set of evaluations the authors used to claim superiority of their method. However in the supplement the authors disclose that they set the default parameters for Rocker-meth based specifically on how it did on the synthetic data from these evaluations. It is thus not clear how meaningful the comparisons are as the authors method got the benefit of this parameter tuning on the evaluation data, but the other methods did not.

I am also concerned that Rocker-meth is reporting statistically invalid p-values/FDRs for DMR regions. The problem is that the first step at the site level has access to the class labels, which can bias the statistical testing in the third step at the region level. The authors can empirically test this by shuffling the class labels and then apply Rocker-meth. If the p-value calculations were valid then they should be uniform on shuffled data, but I would expect the distribution of p-values will deviate from that.

Additional points

- Different gaussians can have different variances in addition to different means. Does this lead to the counter-intuitive situation where higher beta values could be preferred by a Gaussian with lower mean and vice versa?
- Methods state "in cases for which $madj(\beta_{jz}) = 0$, we set $madj(\beta_{jz}) = \max(madj(\beta_{jz}), 1)$ ". As written it seems the expression on right could be simplified to just 1. Also why set it all the way to 1 in such cases, but for other extremely small values it stays at those values?
- "Rocker-meth, allowed for the unbiased detection of DMRs spanning a range of genomic sizes, from few Kilobase-pairs (Kbp) to Megabase-pairs (Mbp)." – Is a few kb actually the minimum detection'?
- 'AUC segments with WMW-value (FDR) ≥ 0.05 or comprising less than 6 data points were discarded for downstream analysis' - What is the justification for discarding segments with less than 6 data points even if the FDR for them is less than 5%
- "We applied Rocker-meth to a wide set of 5,623 cancer samples across 14 TCGA tumor types data, comparing tumor samples with normal matching tissues (712 adjacent tissue samples, requiring ≥ 10 normal samples per tumor type)." – specify in this sentence if this all from array data if that is what it is
- "each 1Kb chromHMM segment was assigned to a differentially methylated state if it was completely within a methylation segment." – what about ChromHMM segments shorter than 1kb?
- The authors refer to their chromatin states as coming from ENCODE, but it seems to actually be from Roadmap Epigenomics
- Methods mention 'imputed chromatin states' but it looks like they were using chromatin states

not based on imputation

- “over-representation of structural-related terms, including Heterochromatin (Het) and Quiescent (Quies) states” – Unclear why these are considered ‘structural’ terms and which terms specifically the authors are considering to be ‘structural’ or not
- A reference to prior work describing/using ‘heterogenous hidden Markov model’ when mentioning that would be appropriate
- I would discourage introducing an entirely new term (‘Hypoway’) to refer to loss of DMR in regions ≥ 10 KB.
- For the abbreviation of ‘bisulfite sequencing’ consider ‘BS-seq’ instead of ‘BS’
- The grammar/wording/typo issues can be improved in a number of places. Here are some examples
 - “curve analyzer of DNA”,
 - “A median the across the datasets”,
 - “Based on these observations on WGBS data, thus well representing the entire genome space,”
 - “for barely all tumor types”
 - “in line with previous report”
 - “Interestingly, we found a relevant number of DMRs shared in more than half of the tumor types”
 - “those results depict a peculiar association between cancer DNA hypermethylation”
 - ‘0,6’ (supplement page 2)
 - “Figure R39” (supplement)
 - “based o the” (supplement)

Reviewer #1 (Remarks to the Author):

In this manuscript, Benelli et al. develop a new method called Rocker-meth for identifying differentially methylated regions in cancer. Rocker-meth uses an AUC segmentation to detect region boundaries, and is particularly robust to noises. Furthermore, the method is applicable to both array and bisulfide sequencing platforms. The authors demonstrate the performance of Rocker-meth with both simulation datasets and TCGA datasets.

A big part of the work is a pan-cancer analysis of DNA methylation profiles. By dividing differentially methylated regions into hypo, hyper, and hypoway, they show cancer genomes harbor substantial hypomethylations, particularly in intergenic regions. In addition, they demonstrate the associations between DMRs and genetic features, and similarly the association between methylation and transcriptional readouts. An interesting observation is that many of the DMRs are shared across cancer types. Finally, they analyzed single-cell methylation data with the new tool and demonstrate the tool can work well for single-cell datasets.

Overall, the manuscript is well written. While the pan-cancer analysis is largely confirmatory, the results lend further validation in addition to the benchmark experiments. I think the paper is of high quality, and thus, endorse its publication.

We would like to thank the Reviewer for the positive feedback and for acknowledging the large body of work we dedicated to this study. In the following, we would like to address his/her suggestions to improve our manuscript. We would like to point out that we changed the term “hypoway” to “hypo-block” to better contextualize our findings compared to the previous literature.

I have the following comments. The authors show high correlation between array and sequencing-based results in Fig S10. Notably, they identified many more hypomethylation events with sequencing data than array data (31% vs 11.5% of the cancer genome). This analysis was not done for hypermethylation. Is it because 450k arrays do not have sufficient coverage in intergenic regions that caused this disparity? For whatever reason, it would be helpful to discuss and offer some insights.

The Reviewer correctly points out the discrepancy of coverage in intergenic regions between sequencing-based techniques (RRBS, WGBS) and microarrays. Indeed, HM450 has a strong gene-centered design and does not cover a great portion of distal regulatory elements or non-regulatory regions. Even though we demonstrated that methylome segmentation greatly mitigates this inherent limitation of HM450, we will inevitably be unable to observe intergenic DMRs that lie exclusively between the sparse HM450 probes. This is probably the case for large intergenic regions, and likely explains the discrepancy observed.

Figure R1. Scatter plot of the burden of genome under hyper-DMRs detected by Rocker-meth in WGBS (y-axis) versus Illumina HM450 across 6 tumor types.

As the Reviewer suggested, we checked for correlation also in hypermethylated regions. Even though the correlation of hyper-DMR burden is not significant, the order of magnitude of differentially methylated genome estimated using the two strategies is comparable. Importantly, the fraction of hypermethylated genome is far smaller than the hypomethylated counterpart, making it more prone to fluctuations dictated by a different strategy. Given that those techniques are extremely different in the number of CpGs covered, we think that this is evidence of the good performance of Rocker-meth segmentation. In addition, we already reported that the CpG island methylator phenotype, a relevant subtype in LUAD, is under-represented in the WGBS dataset, thus likely causing this observed discrepancy.

Based on this analysis, we made the following changes to the text:

- Complemented Supplementary **Figure S10**, showing correlation between hyper-DMRs burden in WGBS vs. HM450.

- Added the following text to the "Results" section:

For hyper-DMRs we observed less correlated signal but still comparable order of magnitude of differentially methylated genome estimated using the two strategies, probably due to the smaller fraction of hypermethylated genome than the hypomethylated counterpart, making it more prone to fluctuations dictated by a different platform or different sample size (Figure S10B).

The second issue is about the single sample analysis. Rocker-meth identifies DMR boundaries based on group comparisons (tumor vs normal), and assumes the boundary applies to each sample when evaluating single tumor samples. It is unclear, however, how heterogeneous the tumor samples are with regard to hypo or hyper-methylated regions. They could share a fraction of the segment but not entirely, in a similar scenario to copy number events. They could be particularly relevant if the targets of methylation events are positively selected during cancer evolution. It is thus important to assess how Rocker-meth performs in this aspect.

We thank the Reviewer for pointing this out. Since Rocker-meth is designed to identify differential DNA-methylation signal between two groups, we are aware that our method could fail in identifying the correct breakpoints in single sample analysis. To evaluate the extent to which this occurs, we first studied the distribution of delta beta (i.e., the difference between the beta of tumor samples and the average beta of normal samples) around the estimated start and end breakpoints for DMRs detected in PRAD. Results are summarized in **Figure R2** showing delta beta for hyper and hypo DMRs detected in PRAD. Overall, we observed that the majority of tumor samples show a coherent signal of differential methylation around the breakpoints detected by Rocker-meth, suggesting their reliable estimations.

Beyond this “qualitative” inspection of DMRs boundaries, we performed a systematic assessment on how Rocker-meth performs in detecting recurrent breakpoints. To do that, we considered the PRAD dataset and perform $n=50$ random sampling of $n=50$ tumor samples (~10% of the dataset) fixing the set of control samples ($n=50$, all the available samples). For each sampling, we applied the method and studied the ability of Rocker-meth in detecting recurrent start and end breakpoint positions for common differentially methylated segments. We then assessed the recurrence of segments’ start and end breakpoints, either alone or in combination. The results are reported in **Figure R3** for hyper and hypo events (without distinction based on length). We observed a remarkable over-representation for shared segments across iterations with ~70% (Hyper) and ~60% (Hypo) of segments boundaries consistent in at least half of iterations. This suggests that most of the events detected by Rocker-meth have common boundaries. This finding supports the reliability of Rocker-meth in estimating the correct breakpoints and provides additional evidence that these segments are markedly conserved across tumor samples.

Figure R2. Box plots of the distributions of the differences of beta in tumor samples vs average beta in normal samples around all starts and ends breakpoints for all hyper (top panel) and hypo+hypo block (bottom panel) DMRs detected in the TCGA-PRAD dataset.

Figure R3. Histogram representing the recurrence of differentially methylated segments across $n = 50$ random permutation of tumor samples against a fixed set of normal (TCGA-PRAD). For each element in the resulting pool of differentially methylated segments, the frequency it's evaluated across the iterations matching the chromosome and either start position, end position, or both (f: frequency).

Reviewer #2 (Remarks to the Author):

In the manuscript "Cancer methylomes characterization enabled by Rocker-meth" by Benelli, Franceschini, et al. the authors present a computational method, Rocker-meth, for calling differentially methylated regions (DMRs). The authors then apply the approach to TCGA data and perform extensive data analyses to characterize the DMRs they called.

The manuscript has two relatively disjoint parts. One part being bioinformatics methods oriented introducing a new DMR calling procedure and providing a few comparisons with existing approaches. The other part was extensive biological focused analyses of the DMRs including with integration of various other genomic data where the specific choice of method to call DMRs seemed secondary.

We thank the Reviewer for his/her observations and for the time dedicated to the constructive criticism provided here.

To identify gold standards to benchmark differential methylation calling algorithms is challenging as genome-wide ground truth is unknown. Each existing algorithm differs in the statistical framework and modeling of differential methylation; therefore, discrepancies arise in any comparative analysis. We therefore reasoned to proceed as follows. First, to employ synthetic datasets to compare overall detection boundaries of *Rocker-meth* and other five

tools (Metilene, DSS, DMRseq, Bumhunter, DMRcate) and to intentionally select synthetic data previously proposed by other investigators (Jühling et al., 2016) (**Figure 1C**). Next, to perform a spectrum of biological analyses to verify the robustness of the DMRs nominated by Rocker-meth. The largely confirmatory nature of those results goes in that direction. In other words, our extensive analysis of cancer methylomes is instrumental to demonstrate the robustness of the method as many of the observed associations have been previously demonstrated using different tools or specific cancer types. Last, with this work we aimed to provide a unified analysis of the differentially methylated regions in cancer as included in the proposed catalog. Throughout the study, we present results that do significantly extend previous knowledge as in the case of the dependency of DNA hypermethylation and transcription factor deregulation on the baseline chromatin state. Additional analyses were performed as part of this revision and are now added to the main manuscript and supplementary material. We made an effort to contextualize them within our study. Importantly, we have now edited the abstract to better present the study rationale and we changed the title of the manuscript.

I felt the new bioinformatics method received more motivation from the abstract and introduction, but most of the results section and the lengthy supplement were focused on the biological analyses. Since the authors only used their method for the biological analyses, it was not clear that the author's new method actually enabled most of the biological results as suggested by the manuscript title.

As the Reviewer suggested, the original title might be misleading, as pan-cancer analyses can be potentially realized with other tools. The advantages provided by Rocker-meth include high accuracy, fast runtime, unbiased segmentation of the genome, and single scores to *measure* nominated DMRs for each sample. While those features do represent substantial improvements in the ability to assess pan-cancer methylomes as compared to other available tools, we rephrased the title as follows:

Charting differentially methylated regions in cancer with Rocker-meth

In addition, we edited the abstract in order to better highlight the findings of our analysis. It now reads as:

Differentially DNA methylated regions (DMRs) inform on the role of epigenetic changes in cancer. We present Rocker-meth, a new computational method exploiting a heterogeneous hidden Markov model to detect DMRs across multiple experimental platforms. Through an extensive comparative study, we first demonstrate Rocker-meth excellent performance on synthetic data. Its application to more than 6,000 methylation profiles across 14 tumor types provides a comprehensive catalog of tumor type-specific and shared DMRs, and agnostically identifies cancer-related partially methylated domains (PMD). In depth integrative analysis including orthogonal omics shows the enhanced ability of Rocker-meth in recapitulating known associations, further uncovering the pan-cancer relationship between DNA hypermethylation and transcription factor deregulation depending on the baseline chromatin state. Finally, we

prove the catalogue amenable to the study of colorectal cancer single-cell DNA-methylation data.

For these two parts to fit well together in the same manuscript, I think the authors should make a stronger case that biological results they are presenting are actually enabled by their new DMR calling procedure relative to existing methods for the same problem. For which analyses, if any, would different biological conclusions have been reached if an existing DMR calling procedure was used and why are the authors confident they are correct in such cases?

Figure S8, is the only comparison on non-simulated data. The authors are comparing odds-ratio without considering coverage of the predictions. How does the coverage of the different methods compare in these evaluations? If the competing methods have greater coverage then it is difficult to compare the methods just based on the odds ratios.

To assess this Reviewer's concern, we performed additional analyses focused on integrative DNA-methylation / gene expression. As no ground truth exists, we focused on events that are more likely to occur, specifically hyper-methylation in promoter-TSS, 5' UTR, or first intron and under-expression. **Figure R4** (top) shows the number of under-expressed genes with a DMR in/spanning their promoter-TSS, 5' UTR, or first intron as predicted by Rocker-meth (yellow), Bumhunter (green) or DMRcate (orange). We observed that Rocker-meth identified a higher number of these events in 8 out of 12 tumor types compared to Bumhunter and 10 out of 12 tumor types compared to DMRcate. Importantly, we observed that these results were not due to an over-calling of DMRs. In fact, as reported in **Figure R4** (bottom) the number of hyper-DMRs detected by Rocker-meth was always lower or comparable to the number of DMRs identified by Bumhunter or DMRcate. These results clearly suggest that Rocker-meth is able to detect more expected (based on biological knowledge) events across different tumor types than the other state-of-the-art methods (high sensitivity) while keeping the number of detected events low (high specificity). These analyses also address the Reviewer's concern about "*how the coverage of the different methods compares in these evaluations*".

Next, we focused on these events and run functional enrichment analysis using clusterProfiler on the set of Gene Ontology (GO) terms (C5: ontology gene sets from MSigDB Collections version 7.4). First, we observed that the number of significant (FDR<0.05) GO terms from Rocker-meth results was markedly higher than those from Bumhunter or DMRcate in 10 out of 12 tumor types (**Figure R5, top**). In addition, Rocker-meth led to more robust GO terms enrichment, as suggested by significantly lower distribution of FDRs in 6 out of 12 tumor types compared to Bumhunter and 10 out of 12 compared to DMRcate (**Figure R5, bottom**).

These further analyses suggested that Rocker-meth may provide more functionally meaningful results compared to available state-of-the-art methods. Importantly, these additional analyses further demonstrate the reliability of our catalog, a valuable, new resource that could be exploited by the cancer genomics/epigenetics community.

Based on these analyses, we made the following changes to the text:

- added the following text to the "Results" section:

As no ground truth exists, we focused on events that are more likely to occur, specifically hyper-methylation in promoter-TSS, 5' UTR, or first intron and under-expression. **Figure 1D** (top) shows the number of under-expressed genes with a DMR in/spanning their promoter-TSS, 5' UTR, or first intron as predicted by Rocker-meth, Bumhunter or DMRcate. We observed that Rocker-meth identified a higher number of these events in 8 out of 12 tumor types compared to Bumhunter and 10 out of 12 tumor types compared to DMRcate. Importantly, we observed that these results were not due to an over-calling of DMRs. In fact, as reported in **Figure S8A** the number of hyper-DMRs detected by Rocker-meth was always lower or comparable to the number of DMRs identified by Bumhunter or DMRcate. These results clearly suggest that Rocker-meth is able to detect more expected (i.e., based on biological knowledge) events across different tumor types than the other state-of-the-art methods (high sensitivity) while keeping the number of detected events low (high specificity). Next, we focused on these events and run functional enrichment analysis using clusterProfiler on the set of Gene Ontology (GO) terms (C5: ontology gene sets from MSigDB Collections version 7.4). First, we observed that the number of significant (FDR<0.05) GO terms from Rocker-meth results was markedly higher than those from Bumhunter or DMRcate in 10 out of 12 tumor types (**Figure S8B**). In addition, Rocker-meth led to more robust GO terms enrichment, as suggested by significantly lower distribution of FDRs in 6 out of 12 tumor types compared to Bumhunter and 10 out of 12 compared to DMRcate (**Figure S8C**). We then studied the ability of the methods in recapitulating other well-known features of DNA methylation in human cancers. As reported in **Figure S8D**, compared to Bumhunter and DMRcate Rocker-meth predicts higher enrichment of hyper-DMRs in promoters and 5' associated with gene under-expression, higher enrichment of hyper-DMRs versus hypo-DMRs in CpG islands and higher enrichment of hypo-DMRs versus hyper-DMRs in intergenic regions.

- Added the new **Figure 1D** reporting the number of under-expressed genes with a DMR in/spanning their promoter-TSS, 5' UTR, or first intron as predicted by Rocker-meth, Bumhunter or DMRcate.

- Modified supplementary **Figure S8** adding the new analyses on expected associations between DNA-methylation and gene expression.

Figure R5.

Top panel. Bar plots of the number of significant (FDR<0.05) Gene Ontology (GO) terms resulting from the analysis of hyper-DMRs in promoter-TSS, 5' UTR or first intron of under-expressed genes detected by Rocker-meth (yellow), Bumphunter (green), and DMRcate (orange) across the different tumor types.

Bottom panel. Box plots showing the distribution of $-\log_{10}$ of FDR from the clusterProfiler analysis on the set of Gene Ontology (GO) terms using events detected by Rocker-meth (yellow), Bumphunter (green), and DMRcate (orange) across the different tumor types. Red dots indicate when no significant term has been found.

Also do the authors have an explanation for why both existing methods compared to are going in the opposite direction for the 'Under-expression/Hyper in promoter/TSS' category

We thank the Reviewer for noticing this and we agree that it is an unexpected result. We checked the Odds-Ratio (OR) related to under-expression and hyper-DMRs in promoter-TSS estimated in each tumor type. The results are reported in the bar plots of **Figure R6 (top)** and indicate low ORs for DMRcate while Bumphunter obtained low ORs for UCEC and PAAD. To understand if these specific results could explain the overall low OR observed for Bumphunter and possibly DMRcate, we run OR analysis excluding these two tumor types from the calculus of the overall OR. As reported in **Figure R6 (bottom)**, Bumphunter obtained very good performance for the class under-expr/hyper-DMRs in promoter-TSS (though still inferior to Rocker-meth) while no relevant difference was observed for DMRcate.

I am concerned about the potential sensitivity of the method to the settings of a number of arbitrary parameters and that the evaluations relative to other methods did not adequately consider that. In particular in the "Comparison study on synthetic data" in the methods section the authors state that the default parameters for all methods were used, which were the main set of evaluations the authors used to claim superiority of their method. However, in the supplement the authors disclose that they set the default parameters for Rocker-meth based specifically on how it did on the synthetic data from these evaluations. It is thus not clear how meaningful the comparisons are as the authors method got the benefit of this parameter tuning on the evaluation data, but the other methods did not.

Based on this Reviewer's comment, we realized that in the original submission we did not properly explain how the Rocker-meth parameters were set.

The evaluation of Rocker-meth on synthetic data and its comparison with the other state-of-the-art methods were performed on synthetic datasets (class 1-4) developed by others (Jühling et al., 2016) with the addition of a more challenging signal-to-noise ratio class (class 5). We also considered two different "spatial configurations" emulating WGBS and Illumina 450K platform.

To set Rocker-meth parameters, prior to the comparative analysis with other tools, we performed a comprehensive simulation study only on synthetic RRBS datasets to evaluate the performance of the segmentation algorithm of Rocker-meth while varying the values of its parameters D_{norm} , p , F , and μ (see Supplementary Material). These analyses allowed us to identify the optimal configuration of these parameters.

Next, we applied Rocker-meth to all the other synthetic datasets and compared its performance with the other state-of-the-art methods. In the current version of the manuscript, we have clarified this point, indicating as "parameters training" the RRBS dataset analysis. Nonetheless, the extremely different datasets that were synthetically generated in our study – including high to low signal-to-noise ratio – and different spatial configurations of CpG sites – including sparse (HM450) and uniform (WGBS) data – we observed that Rocker-meth performs better than the other methods. Of note, Rocker-meth was applied to these datasets using the same set of parameters (default, from RRBS synthetic study) and not adapting its parameters to the characteristics of every single synthetic dataset.

Based on these considerations and regarding the Reviewer's point "... *authors method got the benefit of this parameter tuning on the evaluation data, but the other methods did not*", we still believe that the presented evaluations are reliable and meaningful and that the misunderstanding was caused by lack of clarity in our original presentation. We think that the optimization of the parameters of the other methods based on the characteristics of every single dataset is out of the scope of this manuscript.

To better clarify this point, we made the following changes to the manuscript:

- added the new section "Study of the parameters of the segmentation algorithm of Rocker-meth" in the Methods:

Study of the parameters of the segmentation algorithm of Rocker-meth

To set Rocker-meth parameters, prior to the comparative analysis with other tools (see next section), we performed a comprehensive simulation study only on synthetic RRBS datasets (parameters' training dataset) to evaluate the performance of the segmentation algorithm of

Rocker-meth while varying the values of its parameters (see Supplementary Text). These analyses allowed us to identify the optimal configuration of these parameters.

- indicated the RRBS dataset as "RRBS dataset (parameters' training)" through the manuscript.

I am also concerned that Rocker-meth is reporting statistically invalid p-values/FDRs for DMR regions. The problem is that the first step at the site level has access to the class labels, which can bias the statistical testing in the third step at the region level. The authors can empirically test this by shuffling the class labels and then apply Rocker-meth. If the p-value calculations were valid then they should be uniform on shuffled data, but I would expect the distribution of p-values will deviate from that.

We thank the Reviewer for this important observation that prompted us to clarify and to further test the strategy we employed to characterize the strength of differential methylation in Rocker-meth DMRs.

First, it is worth noting that, in the context of segmentation procedures, obtaining a well-behaved p-value is challenging, as it requires full decoupling between the definition of features and the statistical testing of their signal. Indeed, other DMR calling algorithms (such as DMRcate and bumhunter) deal with this problem using a plethora of different strategies.

As reported by the Reviewer, the AUC step has access to the original labels, which in turn are used to define differentially methylated segments. Consequently, testing DMRs for the difference in beta values across samples would be somehow redundant, as those segments have already been selected to satisfy this requirement. Indeed, in the early stage of the development we verified that merely comparing the averaged beta values within DMRs across groups leads to strong statistical significance for virtually all non-neutral segments, exactly as expected.

For this reason, we devised a strategy to quantify the degree of signal homogeneity within regions. The proposed p-value statistic is thus meant to capture the within stability of DMRs by testing the intra-DMR Beta variability. For each DMR, the p-value is computed by comparing the single CpG site signal averaged across samples between the two groups (e.g. tumor and normal samples). Given the same differential signal, this strategy prioritizes DMRs in which the Beta value is stable, as we performed an unpaired WMW test (toy examples are reported in **Figure R7**). This choice is grounded upon the observation that differential methylation in cancer generally involves a shift from totally methylated or unmethylated states toward less extreme values (Witte, Plass and Gerhauser GenomeMedicine 2014, Landau et al. CancerCell 2014, Berman et al. NatGen2011).

Next, to thoroughly check if Rocker-meth identifies DMRs due to information leakage between steps, we performed a series of permutation experiments ($n = 100$) on the PRAD dataset as suggested by the Reviewer. Briefly, for each iteration, we sampled all the normal samples and the same number of tumors, randomly. In the first 50 iterations, we ran Rocker-meth keeping the correct labels (**Original group**), while in the remaining 50 iterations we shuffled the labels (**Shuffled group**).

First, we observed that for shuffled iterations the number of significant DMRs ($q\text{-value} < 0.05$, $n\text{sites} > 5$) is zero in most cases, with rare exceptions (**Figure R8**). Importantly, the number of DMRs obtained in the correctly labeled iterations display some changes, but the order of magnitude of events is overall coherent across repetitions. Those observations likely rule out the possibility that Rocker-meth is overly prone to false positives due to some sort of overfitting. It is important to notice that the required direct proximity of multiple CpG sites ($n > 5$) with a coherent signal, in combination with the distance-aware HMM used, imposes strong restrictions on the segments that are nominated.

Next, we checked whether the p-values reported by Rocker-meth are overly optimistic (i.e. inflated toward 0). We analyzed the distribution of p-values across all the segments obtained in the above-mentioned permutations (including neutral ones, keeping only segments with $n\text{sites} > 5$).

Figure R9. Density distribution of p-values for all segments obtained with original or shuffled labels. P-values computed on neutral regions are included for both groups.

As expected, we observed that in the shuffled permutations the p-value distribution is clearly different from the one obtained with original labels (**Figure R9**). However, given the nature of our algorithm, it is not possible to observe a uniform distribution under the null hypothesis, as correctly pointed by the Reviewer. Importantly, the distributions of p-values under shuffling are inflated toward 1, making our estimation of robustness particularly conservative. This phenomenon, often termed super-uniformity of p-values, doesn't invalidate the FDR procedure but makes our statistic more conservative (Blanchard and Roquain, *Electron. J. Statist* 2008, Definition 2.1).

We hope to have clarified the above points and we apologize to the Reviewer as in the original submission the testing procedure was ambiguously presented in the method section. Based

on the Reviewer's comments and on the additional analyses, we have made the following changes to the manuscript:

- We modified Figure 1A to clarify this procedure.
- We clarified the implemented p-value computation strategy, corrected the inaccurate terminology (we used intra-segment homogeneity instead of statistical significance) and changed corresponding subsection in Methods section:

Intra-segment homogeneity (step 3). For each AUC segment, intra-segment homogeneity is assessed through the Wilcoxon-Mann-Whitney test between the CpG-wise averaged beta values in cancer samples and normal samples. Resulting p-values are then adjusted by FDR (i.e., Benjamini-Hochberg procedure).

- We also replaced the term q-value with FDR or adjusted p-value through the manuscript, as we employed Benjamini-Hochberg correction.

Additional points:

- *Different gaussians can have different variances in addition to different means. Does this lead to the counter-intuitive situation where higher beta values could be preferred by a Gaussian with lower mean and vice versa?*

We hope to have properly understood the Reviewer's question; we assume the Reviewer was referring to AUC and not to beta values (Rocker-meth performs segmentation of AUC and not beta values). Because of the spatial correlation that the HMM imposes to each state, it is extremely unlikely that, for instance, a hyper-methylated segment (constituted by many consecutive CpGs with high AUC values) is classified as a not-differentially methylated state. On the contrary, if high AUC values belong to the not-differentially methylated state, these will be classified as outliers. In addition, the parameter that governs the split of variance between neutral and differential AUCs has been optimized in the RBBS data based synthetic study and should avoid counterintuitive situations as those posed by the Reviewer.

- *Methods state "in cases for which (MAD lower bound on z-score formula). As written it seems the expression on right could be simplified to just 1. Also why set it all the way to 1 in such cases, but for other extremely small values it stays at those values?*

We thank the Reviewer for bringing this to our attention. Indeed, the Rocker-meth implementation sets the MAD to 1 when the computed value is below 1 as suggested. This choice was dictated by the previous rounding on Beta values and the intrinsic noise of the measurement (i.e. if no variability is observed, we still consider a minimal amount of undetected variability to be present).

We corrected the sentence which now reads as follows:

“[...] to avoid over-estimation of RockerSS in cases for which $mad_j(\overline{\beta_{jz}}) < 1$ (a single percentage point of DNA methylation), we set $mad_j(\overline{\beta_{jz}}) = 1$.”

- “Rocker-meth, allowed for the unbiased detection of DMRs spanning a range of genomic sizes, from few Kilobase-pairs (Kbp) to Megabase-pairs (Mbp).” – Is a few kb actually the minimum detection?

In principle, there is no hard boundary on the length of DMRs as this depends on the density of CpG sites. This is apparent in the WGBS analysis, where the high quality and CpG coverage of the data allowed us to define many DMRs below 1Kbp.

The sentence has been corrected and now reads as follows:

from hundreds of basepairs to Megabase-pair (Mbp) scale.

- ‘AUC segments with WMW-value (FDR) ≥ 0.05 or comprising less than 6 data points were discarded for downstream analysis’ - What is the justification for discarding segments with less than 6 data points even if the FDR for them is less than 5%

The above point raised by the Reviewer represents a recurrent challenge in the context of differential methylation analysis. As previously reported also by Jaffe et al. (PMID: 22422453), considering even a single CpG site as a DMR might capture influential events near TFBS and TSS, but could also lead to a large number of false positives. The arbitrary definition of a DMR is a common issue among all the DMR tools which usually require a user-defined choice of band-width, or delta beta, or max distance to identify these regions. We developed Rocker-meth with the intent to build, and provide a tool to build, catalogs of DMRs that could both be used for specific biological questions and for the interrogation of single samples. We proposed a method to identify genomic segments showing consistent differential methylation signals. In order to compile a robust catalog of these regions, we decided to use a threshold in terms of the number of CpG sites supporting the region for defining a DMR.

To define the threshold of the number of sites for nominating reliable DMRs, we originally studied the distribution of the number of sites supporting false-positive segments detected by Rocker-meth in the synthetic datasets included in this work. Overall, we had observed that false-positive segments tend to be supported by a lower number of sites for datasets characterized by higher noise (classes 4 and 5 vs 1-3), as expected. For HM450, we had obtained no false-positive segment supported by more than 5 sites (the same threshold that we had used to define the catalog) and the same value also corresponded to the fourth quartile of the distribution for WGBS data. We had therefore adopted $n_{sites} = 6$ as a reasonable threshold for calling reliable DMRs in both HM450 and BS-seq real data. The results of this analysis are now included in the manuscript and reported in **Figure R10**.

In addition, we here exploited original and shuffled TCGA-PRAD data reported in the previous response to this Reviewer and looked at the distribution of the number of sites in that setting (**Figure R11**). As expected, we observed that false-positive segments are usually supported by few sites. Importantly, we can also appreciate how $n_{sites} > 5$ is a reasonable threshold to recover *bona fide* DMRs while minimizing false positives.

Importantly, this parameter can be controlled by the user, who can decide to consider segments supported by a lower number of CpG sites for more exploratory applications or in the case of higher resolution platform data.

The study of the number of sites in synthetic and real datasets has now been included in the current version of the manuscript. We made the following changes to the manuscript:

- Added the section "Minimum number of sites for defining reliable DMRs in supplementary text describing the study of the number of sites in false-positive segments;
- Added Supplementary **Figure S4** bottom panel showing the distribution of the number of sites in false-positive segments in HM450, RRBS, and WGBS synthetic datasets.

Figure R10. Box plots of the distribution of the number of false positive segments detected by Rocker-meth in the HM450, RRBS and WGBS synthetic datasets. For high signal-to-noise ratio RRBS datasets (classes 1/2/3) Rocker-meth identified one false positive segment.

Figure R11. Boxplot reporting the number of non-neutral segments detected for all iterations in the sample permutation test, comparing runs with original and shuffled labels. The threshold $n = 6$ is reported as dashed line.

- *“We applied Rocker-meth to a wide set of 5,623 cancer samples across 14 TCGA tumor types data, comparing tumor samples with normal matching tissues (712 adjacent tissue samples, requiring ≥ 10 normal samples per tumor type).” – specify in this sentence if this all from array data if that is what it is*

We added this clarification as suggested. The sentence now reads:

We applied Rocker-meth to a wide set of 5,623 cancer samples across 14 TCGA tumor types profiled with HM450 array, comparing tumor samples with normal matching tissues

- *“each 1Kb chromHMM segment was assigned to a differentially methylated state if it was completely within a methylation segment.” – what about ChromHMM segments shorter than 1kb?*

We thank the Reviewer for noticing this. Indeed, we verified that the resolution of ChromHMM segmentation is 200bp, and not 1Kb. In addition, consecutive segments in the same state are merged together, so referring to the segmentation resolution might cause confusion in the reader. We solved this ambiguity in the sentence that now reads as follows:

As a result, each chromHMM segment was assigned to a differentially methylated state if it was completely within a methylation segment.

- *The authors refer to their chromatin states as coming from ENCODE, but it seems to actually be from Roadmap Epigenomics*

We referred to ENCODE chromatin states as they were first introduced in Kundaje et al. Nature 2015. However, as correctly pointed by the Reviewer, the data itself is from the Roadmap Epigenomics consortium. We clarify this point in the manuscript as follows:

Inferred chromatin states for the hg19 genome were downloaded from (<https://egg2.wustl.edu/roadmap/data/byFileType/chromhmmSegmentations/ChmmModels/coreMarks/jointModel/final/>, Roadmap Epigenomics consortium)

- *Methods mention ‘imputed chromatin states’ but it looks like they were using chromatin states not based on imputation*

In this context, we originally used the term “imputation” not as “imputing missing values”, but as inference of the latent chromatin state that is present within a given region and can be deduced from measured histone marks. To solve this ambiguity, we substituted the term “imputed” with “inferred” in the sentence reported above.

- *“over-representation of structural-related terms, including Heterochromatin (Het) and Quiescent (Quies) states” – Unclear why these are considered ‘structural’ terms and which terms specifically the authors are considering to be ‘structural’ or not*

Based on this reviewer suggestion, and in line with previously reported terminology used in the original paper (PMID: 25693563, Nature 2015) we edited the sentence as:

"...of inactive elements, including Heterochromatin (Het) and Quiescent (Quies) states,[...]"

- *A reference to prior work describing/using 'heterogenous hidden Markov model' when mentioning that would be appropriate*

In the original submission we included Ref. 73 as an example of heterogeneous HMM applied to genomic data. Following the Reviewer's suggestion, we looked for additional references that might be suitable in the context of this manuscript, but we failed to find any. The Reviewer might have suggestions in this direction for us to include.

- *I would discourage introducing an entirely new term ('Hypoway') to refer to loss of DMR in regions ≥ 10 KB.*

The concept of "Hypoways", as originally stated in our manuscript, is very similar to other previous findings (for example partially methylated domains, or hypo-blocks). We now came to the conclusion that introducing a new term for those elements might cause confusion to the reader. To avoid this, we renamed "hypoways" regions into hypo-blocks throughout the whole manuscript. Those entities have been described in multiple papers as large domains displaying usually mild demethylation (PMID: 25191524, PMID: 21706001). We hope that 'hypo-blocks' will allow for a better contextualization of our findings within the current literature.

- *For the abbreviation of 'bisulfite sequencing' consider 'BS-seq' instead of 'BS'*

- We changed it accordingly in the manuscript.

- *The grammar/wording/typo issues can be improved in a number of places. Here are some examples*

"curve analyzer of DNA",

Changed to: *"Receiver operating characteristic curves analyzer for DNA methylation data"*

"A median the across the datasets",

Corrected

"Based on these observations on WGBS data, thus well representing the entire genome space,"

Corrected

“for barely all tumor types”

Corrected

“in line with previous report”

Corrected

“Interestingly, we found a relevant number of DMRs shared in more than half of the tumor types”

Corrected

“those results depict a peculiar association between cancer DNA hypermethylation”

Corrected

‘0,6’ (supplement page 2)

Corrected

“Figure R39” (supplement)

Corrected

“based o the” (supplement)

Corrected

We thank the Reviewer for the attention dedicated to our manuscript. We corrected all typos and improved the clarity of those sentences.

REVIEWERS' COMMENTS:

Reviewer #1 (Remarks to the Author):

The authors have addressed my questions.

Reviewer #2 (Remarks to the Author):

The authors largely addressed my previous comments.

To clarify on two points of confusion from my original review:

In terms about the question on reference to heterogenous Hidden Markov Model. Ref. 73 is fine. My point is it should be cited earlier in the text when heterogenous Hidden Markov Model is first mentioned on p. 3 of the results. Currently Ref. 73 is not cited until p. 16 and it is not explicit it is a reference for a heterogenous Hidden Markov Model from the sentence it appears in: 'In order to identify differentially methylated regions (DMRs), we exploited a strategy inspired by our previous work on the analysis of Copy Number Alterations and runs of homozygosity [73,74].'

In terms of the point with the gaussian's with different variance, the authors are correct I meant the AUC values in my review. At least in theory it seems to me some of the segments with the most extreme AUC values could still get assigned to the Gaussian with a lower mean that has higher variance. The authors give arguments why in practice this might not be a substantial issue though their response did not directly evaluate this.

Additional points

In the formulas on p.18 with the mad_j and beta's being expressed as percentages I would add a '%' after the 1s in the formulas, or express them consistently as non-percentages, since elsewhere beta's are defined to be between 0 and 1 without percentages.

Last sentence of abstract 'Finally, we prove the catalogue amenable to the study of colorectal cancer single-cell DNA-methylation data.' – The grammar of this sentence could be improved. Also 'demonstrate' might be a better word than 'prove'

In terms about the question on reference to heterogenous Hidden Markov Model. Ref. 73 is fine. My point is it should be cited earlier in the text when heterogenous Hidden Markov Model is first mentioned on p. 3 of the results. Currently Ref. 73 is not cited until p. 16 and it is not explicit it is a reference for a heterogenous Hidden Markov Model from the sentence it appears in:

'In order to identify differentially methylated regions (DMRs), we exploited a strategy inspired by our previous work on the analysis of Copy Number Alterations and runs of homozygosity [73,74].'

R. We moved the reference earlier in the text (Result section).

In terms of the point with the gaussian's with different variance, the authors are correct I meant the AUC values in my review. At least in theory it seems to me some of the segments with the most extreme AUC values could still get assigned to the Gaussian with a lower mean that has higher variance. The authors give arguments why in practice this might not be a substantial issue though their response did not directly evaluate this.

R. Based on the Reviewer's comment related to a statement we made in the last rebuttal, we have now checked the distribution of AUCs across the three states within segments identified by the HMM in the PRAD dataset (Figure below). As expected, no extreme AUC values are assigned to the "no differentially methylated" state (gaussian with a lower mean and higher variance), also in case of segments supported by a low number of sites. As this was part of the rebuttal discussion, we did not edit the manuscript.

Figure. Box plots of mean AUC values within segments identified by the HMM of Rocker-meth across ranges of number of segments' sites (1-5, 6-10, 11-20, >20) and of differential methylation states (hypo, neutral, hyper).

In the formulas on p.18 with the μ_j and β 's being expressed as percentages I would add a '%' after the 1s in the formulas, or express them consistently as non-percentages, since elsewhere β 's are defined to be between 0 and 1 without percentages.

R. done.

Last sentence of abstract 'Finally, we prove the catalogue amenable to the study of colorectal cancer single-cell DNA-methylation data.' – The grammar of this sentence could be improved. Also 'demonstrate' might be a better word than 'prove'

R. done.